# Unconventional Superconductivity arising from Multipolar Kondo interactions

Adarsh S. Patri and Yong Baek Kim⋆

Department of Physics and Centre for Quantum Materials, University of Toronto,
Toronto, Ontario M5S 1A7, Canada

⋆ ybkim@physics.utoronto.ca

## Abstract

The nature of unconventional superconductivity is intimately linked to the microscopic nature of the pairing interactions. In this work, motivated by cubic heavy fermion compounds with embedded multipolar moments, we theoretically investigate superconducting instabilities instigated by multipolar Kondo interactions. Employing multipolar fluctuations (mediated by RKKY interaction) coupled to conduction electrons via two-channel Kondo and novel multipolar Kondo interactions, we uncover a variety of superconducting states characterized by higher-angular momentum Cooper pairs, $J = 0, 1, 2, 3$. We demonstrate that both odd and even parity pairing functions are possible, regardless of the total angular momentum of the Cooper pairs, which can be traced back to the atypical nature of the multipolar Kondo interaction that intertwines conduction electron spin and orbital degrees of freedom. We determine that different (point-group) irrep classified pairing functions may coexist with each other, with some of them characterized by gapped and point node structures in their corresponding quasiparticle spectra. This work lays the foundation for discovery and classification of superconducting states in rare-earth metallic compounds with multipolar local moments.



# 1   Introduction

The instability of the Fermi liquid state to interactions is at the heart of a wealth of emergent phenomena. Of particular interest is the superconducting transition, wherein an attractive potential leads to the formation of bound electron pair states. The subsequent macroscopic condensation of these Cooper pairs and mass generation for the gauge fields through the Anderson-Higgs mechanism, leads to the eponymous perfect conductivity and expulsion of magnetic fields. In BCS superconductivity, the pairing of opposite spin electrons by phonon-mediated interactions leads to a superconducting ground state characterized by an isotropic (in momentum space) pairing function and a gapped quasiparticle excitation spectrum.

However, the discovery of superconductivity in a variety of strongly-correlated systems – including cuprates [1], heavy fermions [2–15], transition metal oxides [16–23], organic [24–26] and U-based ferromagnetic [27–30] superconductors – has challenged this conventional wisdom. Appropriately named as unconventional superconductors, they have broadly been characterized by anisotropic condensate wavefunctions, odd/even under spatial parity and time-reversal, as well as possessing gapless (nodal) structures in the quasiparticle spectrum [31,32]. Understanding their microscopic origins led to decades of active research, which has given rise to a number of proposed mechanisms that go beyond the phonon-mediated description of conventional superconductors. For instance, magnetic/spin fluctuations have been attributed to the origin of the odd-parity superconductivity in heavy fermion $UPt_3$ [33] and the $d$-wave superconductor $UPd_2Al_3$ [34–36], while orbital fluctuations have been suggested as the cause in the iron-pnictides [37]. The evidently intimate link between the nature of the superconducting state and the interaction that instigates its formation leads one to question if novel superconducting instabilities may occur from novel interactions.

The investigation of rare-earth metallic compounds provides an ideal avenue to explore this question. Through the combination of spin-orbit coupling and crystalline electric fields, the localized rare-earth ions support anisotropic charge and magnetization densities, described by higher-rank multipolar moments [38–41]. As a consequence of their non-trivial transformations under lattice symmetries, conduction electrons may interact with them in atypical manners. For instance, in the single-impurity limit, so-called multipolar Kondo interactions lead to the development of both multi-channel as well as exotic non-Fermi liquid states, where both the conduction electron spin and orbital degrees of freedom become intertwined under scattering events with the moment [42–45]. In the generalized lattice setting, this Kondo effect competes with RKKY interactions between the moments leading to a rich phase diagram of exotic phenomena including hidden multipolar-ordered phases [46–50, 50–53], emergent non-Fermi liquids [54–56], and unconventional superconductivity [57–61] in the neighbourhood of a putative quantum critical point [62,63]. In ferro-quadrupolar $PrTi_2Al_{20}$, for example, thermodynamic and transport measurements indicate the existence of a broad superconducting dome coexisting with ferro-quadrupolar ordering under hydrostatic pressure [64]. Indeed, the $T_c$ is enhanced near the critical point suggesting that multipolar/orbital fluctuations of the local moments play a crucial role in the pairing mechanism. Since the interactions between the moments and electrons may themselves be unusual, this provides the tantalizing prospect of the development of exotic unconventional superconducting behaviours.

In this work, motivated by superconducting behaviours in ferro-quadrupolar $PrTi_2Al_{20}$ [64], we investigate the superconducting instability instigated by multipolar Kondo interactions. Employing a Ginzburg-Landau theory of ferro-multipolar ordering (mediated by RKKY interaction), we consider Gaussian multipolar fluctuations in the high-symmetry paramagnetic phase. These fluctuations (and the associated order parameters) are symmetry-permitting and coupled to conduction electrons possessing spin $(\uparrow, \downarrow)$ and orbital $(\ell = 1)$ degrees of freedom. Due to the nature of the Kondo coupling, the electrons' decoupled spin and orbitals are interwoven to form effective $j = \frac{1}{2}, \frac{3}{2}$ conduction electrons. The multipolar Kondo interaction used in this work was recently shown to result in both two-channel and novel non-Fermi liquid behaviours in the single-impurity limit; as such, we consider these two limiting Kondo interactions as the source of superconductivity. Integrating out the Gaussian multipolar fluctuations, and employing group theoretical methods, we derive the superconducting interaction wherein the Cooper pair channels are organized into the $(O_h)$ cubic symmetry irreps, $A_{1g}, A_{2g}, T_{1g}, T_{2g}$, which involve combinations of the total angular momentum of the Cooper pairs, $J = 0, 1, 2, 3$; for brevity we drop the $g$erade subscript henceforth.

The pairing functions arising from the two-channel Kondo interaction, have even/odd spatial parity that follows from their even/odd $J$. Intriguingly, from the novel Kondo interaction, both even/odd parity channels are possible regardless of the Cooper pair's total angular momentum. This is a marked difference from conventional BCS and highlights the exotic nature introduced from the novel Kondo interaction. Using mean-field theory, we examine the corresponding quasiparticle spectra and discover either point nodes along the various cubic axes [100], [110] and [111], or a fully gapped spectrum on the Fermi surface (with a momentum space dependence acquired from the pairing potential) depending on the superconducting irrep of interest. This work lays the foundation for the discovery of unusual forms of superconductivity in multipolar based heavy fermion compounds.

The rest of the paper is organized as follows. In Sec. 2 we present a Ginzburg-Landau theory of multipolar fluctuations based on the symmetry-constraining environment surrounding the multipolar moments. In Sec. 3 we consider the multipolar Kondo coupling of conduction electrons (of total angular momentum $j = \frac{1}{2}, \frac{3}{2}$) to ferro-multipolar order parameters. The Gaussian multipolar fluctuations are then integrated out to obtain effective electron-electron interactions that can instigate superconducting instabilities. Section 4 organizes the subse-

quent pairing Hamiltonian (composed of Cooper pair operators of total angular momentum $J \in [0,3]$) into the irreducible representations of the $O_h$ point group. In Sec. 5 we consider the variety of superconducting order parameters arising from two-channel and novel multipolar Kondo interactions, and discuss the variety of different pairing irreps. Section 6 details the properties of the non-trivial superconducting states (including the nodal structure of the quasiparticle spectra) using a mean-field theory approach. Lastly, in Sec. 7 we discuss the key findings from this study and provide directions of future work.

## 2 Ginzburg-Landau theory of multipolar ordering

Localized multipolar moments arise from a combination of spin-orbit coupling and crystalline electric field (CEF) effects. As a representative example, we consider the family of cubic multipolar compounds, $Pr(Ti,V)_2Al_{20}$, where the Pr ions reside on a two-sublattice diamond lattice. Encircling each Pr ion is a cage of Al-atoms that subjects the $4f^2$ electrons to a local $T_d$ symmetry, which splits the $J = 4$ multiplet to yield a low-lying non-Kramers doublet of ground states, $|\Gamma_3^{1,2}\rangle$. These states support solely higher-rank multipolar moments, namely time-reversal even quadrupolar moments $\hat{\mathcal{O}}_{20} = \frac{1}{2}(3J_z^2 - J^2)$, $\hat{\mathcal{O}}_{22} = \frac{\sqrt{3}}{2}(J_x^2 - J_y^2)$, and a time-reversal odd octupolar moment $\hat{\mathcal{T}}_{xyz} = \frac{\sqrt{15}}{6}\overline{J_xJ_yJ_z}$ [41]. The two-fold nature of the ground state permits a tidy representation of the multipolar moments in terms of pseudospin-1/2 operators $\mathbf{S} = (S^x, S^y, S^z)$,

$$S_{A,B}^x \sim \hat{\mathcal{O}}_{22}, \qquad S_{A,B}^y \sim \hat{\mathcal{O}}_{20}, \qquad S_{A,B}^z \sim \hat{\mathcal{T}}_{xyz}. \tag{1}$$

Due to the sublattice nature of the underlying diamond structure, the local moments are also specified by their sublattice (A,B) location.

Conduction electron mediating RKKY-like interactions permit the development of spontaneous ferro- and antiferro- multipolar orderings. In this work, we focus on the possible development of ferro-like order of both quadrupolar and octupolar moments described by the coarse-grained Landau order parameters [65],

$$\phi_{x,y,z}(\mathbf{r}) = \langle S_A^{x,y,z}(\mathbf{r}) \rangle + \langle S_B^{x,y,z}(\mathbf{r}) \rangle, \tag{2}$$

where $\mathbf{r}$ denotes the coarse-grained spatial coordinate. We note that in the subsequent path-integral formulation, $\phi_{x,y,z}$ are bosonic field variables.

Constrained by the surrounding $T_d$ point group of each moment, time-reversal symmetry, and spatial inversion about bond-centre (detailed in Appendix A), we have the following Euclidean time static Ginzburg-Landau action for the multipolar moments,

$$S_0 = \int_\tau \sum_{\mathbf{q}} \sum_{\mu\nu} \phi_\mu(-\mathbf{q})\mathcal{M}_{\mu\nu}(\mathbf{q})\phi_\nu(\mathbf{q}), \tag{3}$$

where we employ the Fourier modes of the order parameters, $\int_\tau = \int_0^\beta d\tau$, $\mathcal{M}_{xx}(\mathbf{q}) = (m_{\mathcal{Q}} + a_0\mathbf{q}^2 + a_2q_\nu^2)$, $\mathcal{M}_{yy}(\mathbf{q}) = (m_{\mathcal{Q}} + a_0\mathbf{q}^2 - a_2q_\nu^2)$, $\mathcal{M}_{zz}(\mathbf{q}) = (m_{\mathcal{O}} + a_1\mathbf{q}^2)$, $\mathcal{M}_{xy}(\mathbf{q}) = \mathcal{M}_{yx}(\mathbf{q}) = (a_2q_\mu^2)$. We employ the cubic $E_g$ normal modes $q_\nu^2 \equiv \frac{1}{2}(2q_z^2 - q_x^2 - q_y^2)$ and $q_\mu^2 \equiv \frac{\sqrt{3}}{2}(q_x^2 - q_y^2)$, phenomenological constants $m_{\mathcal{Q},\mathcal{O}}$ represent the mass terms for the quadrupolar and octupolar degrees of freedom, and $a_{0,1,2} > 0$ express the stiffness associated with spatial fluctuations of the order parameters. We retain only the quadratic fluctuations of the order parameters under an implicit Gaussian approximation that only weak fluctuations are important for the superconducting instability in the approach from the paramagnetic phase ($m_{\mathcal{Q},\mathcal{O}} > 0$).

# 3 Electron-electron interactions from multipolar Kondo effects

The nature of the interaction between the multipolar moments and conduction electrons, and the subsequent many-body ground state, is strongly dependent on the available conduction electron spin and orbitals [42–45]. As a representative example, we consider conduction electrons, characterized by orbital $p_{x,y,z}$ ($l = 1$) and spin$-1/2$ degrees of freedom, forming a Fermi surface well localized about the high-symmetry zone-centre of the Brillouin zone [66,67]. The electrons are degenerate in both orbital and spin space, where the orbital degeneracy in the $p$ orbitals satisfies the cubic ($O_h$) symmetry of the high-symmetry zone-centre and the spin degeneracy is guaranteed by time-reversal symmetry. The free fermion action is given by,

$$S_c = \int_\tau \sum_{\mathbf{k}} \overline{c}_{\mathbf{k}\mu} \Big[ (\partial_\tau + \epsilon_{\mathbf{k}}) \delta_{\mu\nu} \Big] c_{\mathbf{k}\nu}, \tag{4}$$

where $\mu, \nu$ run over the six flavours of fermions (Appendix B details the conduction basis used) which have a degenerate dispersion $\epsilon_{\mathbf{k}} = \frac{\mathbf{k}^2}{2m} - \mu_F$.

Though the isolated conduction electrons do not necessarily possess intrinsic spin-orbit coupling, the interaction with multipolar Kondo moments necessitates the intertwining of the orbital and spin degrees of freedom,

$$S_K = \int_\tau \sum_{\mathbf{k},\mathbf{q}} \Big[ \overline{c}_{\mathbf{k}+\mathbf{q},\mu} \Big[ \Gamma^x_{\mu\nu} \phi_x(\mathbf{q}) + \Gamma^y_{\mu\nu} \phi_y(\mathbf{q}) + \Gamma^z_{\mu\nu} \phi_z(\mathbf{q}) \Big] c_{\mathbf{k},\nu} + (\mathbf{q} \leftrightarrow -\mathbf{q}) \Big], \tag{5}$$

where we detail the form of the interaction vertices $\Gamma^{x,y,z}$ in Appendix B; it suffices to state here that the Kondo interaction vertices involve three coupling constants $J_{1,2,3}$. The natural basis for the conduction electrons in Eq. 5 is in terms of the total angular momentum $j = \ell \otimes s = 1 \otimes \frac{1}{2} = \frac{1}{2}, \frac{3}{2}$. In the single impurity limit, this multipolar Kondo interaction permits the development of (i) a two-channel non-Fermi liquid behaviour (characterized by $J_2 = 0$), and (ii) a novel non-Fermi liquid behaviour ($J_1 = 0$) at low temperatures [44].

The Gaussian nature of the multipolar order parameters permits them to be integrated out (as described in Appendix C) of the path integral to yield an effective action,

$$Z = \int \mathcal{D}[\overline{c}, c; \overline{\phi}, \phi] e^{-(S_0 + S_c + S_K)} = \int \mathcal{D}[\overline{c}, c] e^{-S_c} e^{-S_{\text{eff}}}. \tag{6}$$

In order to study the superconducting instabilities instigated by $S_{\text{eff}} = \int_\tau H_{\text{eff}}$, we rewrite it in terms of pairing channel terms by (i) normal ordering the interaction and (ii) projecting the interactions to ensure Cooper pairs are formed from electrons of opposite momenta (in the renormalization group sense, any other possible pairs yield irrelevant interaction vertices [68]). The effective superconducting Hamiltonian is,

$$H_{\text{eff}} = -\sum_{\mathbf{k},\mathbf{k}'} \sum_{\alpha\beta\gamma\delta} (\mathcal{V}_{\alpha\beta\gamma\delta})_{\mathbf{k}-\mathbf{k}'} c^\dagger_{\mathbf{k},\alpha} c^\dagger_{-\mathbf{k},\gamma} c_{-\mathbf{k}',\delta} c_{\mathbf{k}',\beta}, \tag{7}$$

where the complete form of the interaction potential $(\mathcal{V}_{\alpha\beta\gamma\delta})_{\mathbf{k},\mathbf{k}'}$, involving bi-quadratic products of the interaction vertices $\Gamma^{x,y,z}$ and momentum dependent form factors, is presented in Appendix C. We note that the interaction potential is composed of rational functions in momentum $\mathbf{k}, \mathbf{k}'$, which introduces a challenge when attempting to separate it into a product of a function solely dependent on $\mathbf{k}$ and a function solely dependent on $\mathbf{k}'$; this difficulty encourages us to retain this momentum dependence $\mathbf{k}, \mathbf{k}'$ in the interaction/vertex function when numerically solving the gap equations. The abundance of possible terms in Eq. 7 encourages a careful examination of two limiting cases of the effective electron-electron interaction generated from (i) two-channel Kondo interaction, and (ii) novel multipolar Kondo interaction, which we do so in the subsequent sections.

# 4 Higher-angular momentum Cooper Pairs

The nature of the effective interaction permits superconducting instabilities of electrons belonging solely in the $j = 1/2$ sector, solely in the $j = 3/2$ sector, and a mixture of the two $j$ sectors. This leads to the development of Cooper pairs of total angular momentum $J$,

$$\frac{1}{2} \otimes \frac{1}{2} \to 0 \oplus 1 \,, \tag{8}$$

$$\frac{3}{2} \otimes \frac{3}{2} \to 0 \oplus 1 \oplus 2 \oplus 3 \,, \tag{9}$$

$$\frac{1}{2} \otimes \frac{3}{2} \to 1 \oplus 2 \,, \tag{10}$$

which follows from standard angular momentum addition. This higher-angular momentum nature of the Cooper pair is unlike the standard singlet state of BCS theory. We note that the conduction electrons created here possess only the total angular momentum (or "effective spin") $j$ and no additional orbital angular momentum.

Formally, the angular momentum sector of a generic form of a Cooper pair creation operator with momentum $\mathbf{k}$ can be decomposed into the total angular momentum states [69],

$$c^\dagger_{\mathbf{k};j_1} c^\dagger_{-\mathbf{k};j_2} = \sum_{J,M} \langle j_1, m_1; j_2, m_2 | j_1, j_2; J, M \rangle b^\dagger_{J,M;\mathbf{k}} \,, \tag{11}$$

where $j_{1,2}$ and $m_{1,2}$ are the effective spin and $z$-direction component of each electron ($m_{1,2}$ subscripts for the conduction electron operators are dropped for brevity), and $b^\dagger_{J,M;\mathbf{k}}$ is a Cooper pair creation operator of total effective spin $J$ and $M$ component along the $z$-direction at momentum $\pm\mathbf{k}$. $\langle j_1, m_1; j_2, m_2 | j_1, j_2; J, J_z \rangle$ are the Clebsch-Gordon (CG) coefficients, which takes into account the symmetric/anti-symmetric property of effective spin exchange via the phase factor i.e. $\langle j_1, m_1; j_2, m_2 | j_1, j_2; J, M \rangle = (-1)^{J-j_1-j_2} \langle j_2, m_2; j_1, m_1 | j_2, j_1; J, M \rangle$.

Due to the electrons possessing both $j$ and $\mathbf{k}$ quantum numbers, fermionic exchange involves the composition of spin exchange ($j_1 \leftrightarrow j_2$) and spatial parity ($\mathbf{k} \to -\mathbf{k}$). For $j_1 = j_2$, the associated Cooper pair operator must be even (odd) under spatial parity if $J$ is even (odd) to satisfy Fermi-Dirac statistics; this can verily be identified from Eq. 11 using the aforementioned CG phase factor and the anticommutation of fermionic operators. For $j_1 \neq j_2$, special care needs to be taken to establish the spatial parity nature of the pairing operator, as one can define a Cooper pair creation operator in two ways: $b^\dagger_{J,M,\mathbf{k}}$ ($\tilde{b}^\dagger_{J,M,\mathbf{k}}$) with $j = \frac{3}{2}$ ($j = \frac{1}{2}$) fermion at $\mathbf{k}$ and $j = \frac{1}{2}$ ($j = \frac{3}{2}$) fermion in the $-\mathbf{k}$ in Eq. 11. These Cooper pair operators are related to each other by the CG phase factor under exchange of $j_1$ and $j_2$. In order to create a Cooper pair of definite parity, one should thus consider linear combinations ($\pm$) of the Cooper pair operator in Eq. 11,

$$b^\dagger_{J,M;\mathbf{k};\pm} = \frac{1}{\sqrt{2}} \left( b^\dagger_{J,M;\mathbf{k}} \pm \tilde{b}^\dagger_{J,M;\mathbf{k}} \right) \,, \tag{12}$$

where due to satisfying Fermi-Dirac statistics, $\tilde{b}^\dagger_{J,M;-\mathbf{k}} = (-1)^{J+1} b^\dagger_{J,M;\mathbf{k}}$. Thus, $b^\dagger_{J,M;\mathbf{k};+}$ ($b^\dagger_{J,M;\mathbf{k};-}$) is odd (even) under spatial parity for even $J$; and $b^\dagger_{J,M;\mathbf{k};+}$ ($b^\dagger_{J,M;\mathbf{k};-}$) is even (odd) under spatial parity for odd $J$.

The cubic nature of the interactions necessitates that the Cooper pair operators of total angular momentum $b^\dagger_{J,M;\mathbf{k}}$ be organized into the irreducible representations of the associated point-group $O_h$, rather than the good quantum number of spherical symmetry, $J$ [70–72]. The group theoretical decomposition of Cooper pair states are: for $J = 0 \to A_1$,

$$|A_1\rangle = |0,0\rangle \,, \tag{13}$$

for $J = 1 \to T_1$,

$$|T_{1^{(1)}}\rangle = \frac{1}{\sqrt{2}}\Big[ |1,1\rangle - |1,-1\rangle \Big], \tag{14}$$

$$|T_{1^{(2)}}\rangle = \frac{i}{\sqrt{2}}\Big[ |1,1\rangle + |1,-1\rangle \Big], \tag{15}$$

$$|T_{1^{(3)}}\rangle = |1,0\rangle, \tag{16}$$

for $J = 2 \to E \oplus T_2$,

$$|E_{(1)}\rangle = \frac{1}{\sqrt{2}}\Big[ |2,2\rangle + |2,-2\rangle \Big], \tag{17}$$

$$|E_{(2)}\rangle = |2,0\rangle, \tag{18}$$

$$|T_{2^{(1)}}\rangle = \frac{i}{\sqrt{2}}\Big[ |2,1\rangle + |2,-1\rangle \Big], \tag{19}$$

$$|T_{2^{(2)}}\rangle = \frac{1}{\sqrt{2}}\Big[ |2,1\rangle - |2,-1\rangle \Big], \tag{20}$$

$$|T_{2^{(3)}}\rangle = \frac{i}{\sqrt{2}}\Big[ |2,2\rangle - |2,-2\rangle \Big], \tag{21}$$

and for $J = 3 \to A_2 \oplus T_1 \oplus T_2$,

$$|A_2\rangle = \frac{i}{\sqrt{2}}\Big[ |3,2\rangle - |3,-2\rangle \Big], \tag{22}$$

$$|T_{1^{(1)}}\rangle = \sqrt{\frac{5}{16}}\Big[ |3,3\rangle - \sqrt{\frac{3}{5}}|3,1\rangle + \sqrt{\frac{3}{5}}|3,-1\rangle - |3,-3\rangle \Big], \tag{23}$$

$$|T_{1^{(2)}}\rangle = i\sqrt{\frac{5}{16}}\Big[ |3,3\rangle + i\sqrt{\frac{3}{5}}|3,1\rangle + \sqrt{\frac{3}{5}}|3,-1\rangle + |3,-3\rangle \Big], \tag{24}$$

$$|T_{1^{(3)}}\rangle = |3,0\rangle, \tag{25}$$

$$|T_{2^{(1)}}\rangle = \sqrt{\frac{3}{16}}\Big[ |3,3\rangle + \sqrt{\frac{5}{3}}|3,1\rangle - \sqrt{\frac{5}{3}}|3,-1\rangle - |3,-3\rangle \Big], \tag{26}$$

$$|T_{2^{(2)}}\rangle = i\sqrt{\frac{3}{16}}\Big[ |3,3\rangle - \sqrt{\frac{5}{3}}|3,1\rangle - i\sqrt{\frac{5}{3}}|3,-1\rangle + |3,-3\rangle \Big], \tag{27}$$

$$|T_{2^{(3)}}\rangle = \frac{1}{\sqrt{2}}\Big[ |3,2\rangle + |3,-2\rangle \Big], \tag{28}$$

where we use the notation of the irreps of $O_h$ point group, and $|J, M\rangle$ is the total angular momentum wavefunction of the Cooper pair operator $b^\dagger_{J,M;\mathbf{k}}$. We stress that the irrep decomposition is of the total angular momentum $J$, rather than the linear momentum, of the Cooper pair. The above total angular momentum states are even (with the appropriate $g$erade subscript) under the inversion element of $O_h$ due to being composed of orbital angular momentum $l_1 = l_2 = 1$ electrons i.e. picks up a phase of $(-1)^{l_1+l_2} = 1$ under the inversion [73]. We contrast this with the spatial parity that flips the linear momentum $\mathbf{k}$. As mentioned previously, we have dropped the $g$erade subscript for brevity. The procedure by which the group decomposition is performed is detailed in Appendix D, and the composition of the Cooper pair in terms of individual fermionic bilinears is presented in Appendix E. We note that the cubic irrep basis functions and the corresponding Cooper pair operators are time-reversal invariant. This permits the subsequent superconducting states to be characterized as time-reversal preserving (breaking) depending on if the superconducting order parameters, $\Delta$, are (not) equal to their complex conjugate, up to a global phase [70], as is typical in studies of multicomponent superconductivity.

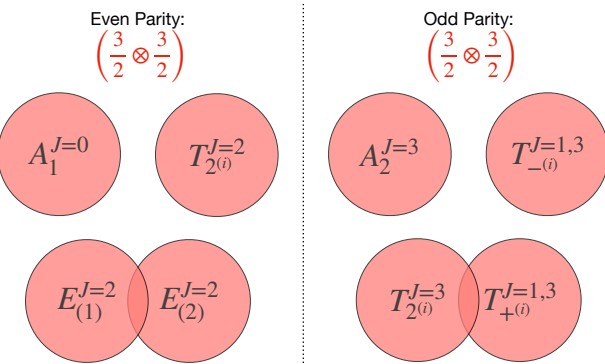

Figure 1: Superconducting order parameters arising from $\frac{3}{2} \otimes \frac{3}{2}$ electrons, with $i = 1, 2, 3$ denoting the components of the three-dimensional irreps, from the two-channel Kondo interaction. Coupled order parameters have an intersection between their depicted circles. The Cooper pair operators associated with the order parameters are detailed in Appendix E. The complete form of the pairing Hamiltonian is presented in Appendix F.

## 5 Superconducting instabilities from multipolar Kondo interactions

The Cooper pairs, and the associated pairing functions, are associated with definite spatial parity. In order to account for this, the interaction potential must similarly be decomposed into even and odd under spatial parity components [74],

$$
(V_\Gamma^\pm)_{\mathbf{k}-\mathbf{k}'} = \frac{1}{2}\Big[ (V_\Gamma)_{\mathbf{k}-\mathbf{k}'} \pm (V_\Gamma)_{\mathbf{k}+\mathbf{k}'} \Big],
\tag{29}
$$

where the $\Gamma$ indicates a particular irrep of interest of definite spatial parity, and the interaction potential is inversion-symmetric $(V_\Gamma)_{\mathbf{k}-\mathbf{k}'} = (V_\Gamma)_{-\mathbf{k}+\mathbf{k}'}$. From inspection, $(V_\Gamma^+)_{\mathbf{k}-\mathbf{k}'}$ and $(V_\Gamma^-)_{\mathbf{k}-\mathbf{k}'}$ are respectively even and odd under spatial parity, and thus the interaction Hamiltonian will project out terms of definite parity i.e. pairing operators even (odd) under spatial parity only contain the associated $V_\Gamma^+$ ($V_\Gamma^-$) portions of the interaction potential.

### 5.1 Two-channel Kondo interaction derived pairing instabilities

The superconducting order parameters derived from the two-channel Kondo interaction involve electrons solely belonging to the $j = \frac{3}{2}$ sector. They can divided into two families: those involving even-$J$ and odd-$J$ total angular momentum, which correspond to even-(odd-)$J$ pairing functions under spatial parity. Since the interaction potential is functionally dependent on momentum space, this permits certain Cooper pairs of different cubic irreps to scatter off each other. For the even-$J$ sector, $A_1^{J=0}$ and $\vec{T}_2^{J=2}$ are decoupled from the rest, while the two components of the two-dimensional $\vec{E}_2^{J=2}$ irrep are numerically found to be non-vanishing. For the odd-$J$ sector, the variety of realized irreps is more prominent as though the $A_2^{J=3}$ sector is decoupled from the rest, the $\vec{T}_1^{J=1}$ and $\vec{T}_1^{J=3}$ irreps form two linear combinations out of which one of the linear combination (for each component of the three dimensional irrep $T_1$) couples to a component of $\vec{T}_2^{J=3}$. The complete form of the Hamiltonian is presented in Appendix F. We present a schematic depicting the decoupling for the even and odd-$J$ channels, and subsequent variety of the irreps in Fig. 1.

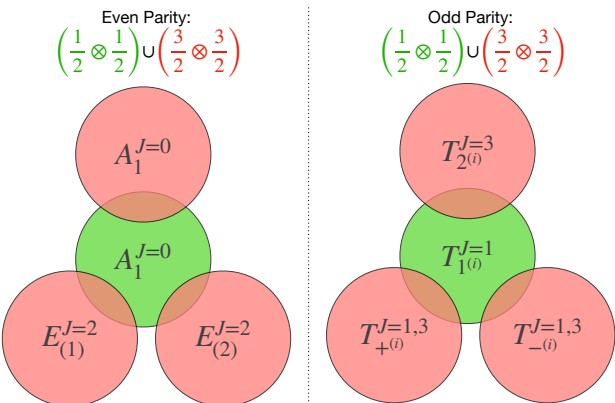

Figure 2: Superconducting order parameters arising from $\frac{3}{2} \otimes \frac{3}{2}$ and $\frac{1}{2} \otimes \frac{1}{2}$ electrons, with $i = 1, 2, 3$ denoting the components of the three-dimensional irreps, from the novel fixed point Kondo interaction. Coupled order parameters have an intersection between their depicted circles. The Cooper pair operators associated with the order parameters are detailed in Appendix E. The complete form of the pairing Hamiltonian is presented in Appendix G.

## 5.2 Novel Kondo interaction derived pairing instabilities

The superconducting order parameters derived from the novel Kondo interaction involve electrons belonging to the $j = \frac{1}{2}$ and $j = \frac{3}{2}$ sectors. This offers a novel avenue of diversity of superconductivity as Cooper pairs may be (i) formed from within each sector separately and scatter off pairs in the other sector, and (ii) may be composed of one fermion from $j = \frac{1}{2}$ and the other from $j = \frac{3}{2}$ sector. The classification of even/odd spatial parity Cooper pairs for scenario (i) follows the previous approach, namely it is identified with the even/odd $J$ nature of the Cooper pair. As detailed in Sec. 4, the parity identification for scenario (ii) is not as simple, and one has both even and odd parity pairings regardless of the even/odd-ness of $J$. This is a clear distinction from the instabilities arising from the two-channel Kondo interaction. We depict the variety of the various pairing channels for scenario (i) and (ii) in Figs. 2 and 3, respectively. As seen in Fig. 2, though the components of the Cooper pairs formed within the $j = \frac{3}{2}$ sector are decoupled from each other (for both even and odd $J$), the Cooper pair formed within with $j = \frac{1}{2}$ sector provides a common source to scatter the decoupled channels amongst themselves. For the scenario (ii) in Fig. 3, even and odd $J$ Cooper pairs are permitted to scatter off each other, since even/odd $J$ no longer corresponds to even/odd spatial parity pairing functions.

## 6 Properties of Superconducting States

The characterization of the superconducting channels in terms of even/odd parity cubic irreps permits a BCS-mean field theory to be developed to study the properties of the superconducting state. Employing a Hubbard–Stratonovich (HS) transformation (as detailed in Appendix H) we obtain gap equations of the form,

$$\Delta_{\mathbf{p}\alpha} = \sum_{\mathbf{q}} \sum_{\gamma} (\Omega_{\mathbf{pq}})_{\alpha\gamma} \sum_{i=1,\ldots,m} \frac{\tanh(\beta E_{m\mathbf{q}}/2)}{E_{m\mathbf{q}}} \frac{\partial E_{i\mathbf{q}}^2}{\partial \overline{\Delta}_{\mathbf{q}\gamma}}, \tag{30}$$

where $E_{m\mathbf{q}}$ is the Bogoliubov quasiparticle dispersion of the $m^{\text{th}}$ quasiparticle, $(\Omega_{\mathbf{pq}})_{\alpha\gamma}$ is a collection of interaction potentials associated with a decoupled irrep (or collection of irreps

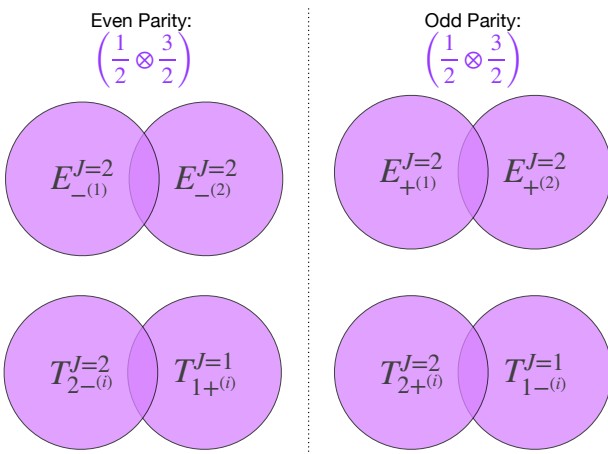

Figure 3: Superconducting order parameters arising from $\frac{1}{2} \otimes \frac{3}{2}$ electrons, with $i = 1, 2, 3$ denoting the components of the three-dimensional irreps, from the novel fixed point Kondo interaction. Coupled order parameters have an intersection between their depicted circles. The Cooper pair operators associated with the order parameters are detailed in Appendix E. The complete form of the pairing Hamiltonian is presented in Appendix G.

that is decoupled from the rest), and $\beta = 1/T$. In the case of a unique ($m = 1$) quasiparticle dispersion, as is for the decoupled pairing channel in $\frac{3}{2} \otimes \frac{3}{2}$ and the pairings from case of (ii) of the novel interaction, there is an additional factor of the 2 on the right hand side of Eq. 30.

For choice of parameters provided in Appendix H, we obtain non-trivial gap solutions for particular irreps. Indeed, solving the coupled BCS gap equations (Eq. 30) yields multiple non-vanishing order parameter solutions in some cases. We present a summary table for the various non-vanishing order parameters in Table 1 and 2 for the two-channel Kondo interaction and the novel multipolar Kondo interaction, respectively. As seen, for the choice of parameters, not all order parameters are realized, and in some cases multiple irrep solutions are found. In Appendix I, we present the **k**-space distribution of the realized order parameters.

The nature of the superconducting state and the accompanying Bogoliubov quasiparticle dispersion is intimately linked to the composition of the Cooper pairs from the fermionic $j$ sectors. For Cooper pairs formed from $j = \frac{3}{2}$ or $j = \frac{1}{2}$ sector, the realized even-parity order parameters have gapped quasiparticle dispersion, though with a momentum space dependence that follows from the momentum distribution of the interaction potential. This property ap-

Table 1: Non-vanishing superconducting states resulting from electron-electron interactions induced by two-channel Kondo coupling. G.D. = Gapped dispersion with acquired $k$ dependence from pairing potential. The corresponding order parameters are provided in Fig. 1.

| Spatial Parity | $\Delta$ | Superconducting Properties |
|---|---|---|
| Even | $A_1^{J=0}$ | G.D. Time-reversal invariant state. |
| | $E^{J=2}$ | G.D. Time-reversal invariant states. |
| Odd | $T_{2(1,2,3)}^{J=3},$ $T_{+(1,2,3)}^{J=1,3}$ | Coexisting superconducting order parameters. Gapless point nodes along body diagonals $[111], [\bar{1}11], [1\bar{1}1], [11\bar{1}]$ axes. Time-reversal broken states. |
| | $T_{-(1,2,3)}^{J=1,3}$ | Gapless point nodes along cubic $[100], [010], [001]$ axes. Time-reversal broken states. |

Table 2: Non-vanishing superconducting states resulting from electron-electron interactions induced by novel Kondo coupling. G.D. = Gapped dispersion with acquired $k$ dependence from pairing potential. The corresponding order parameters are provided in Figs. 2, 3.

| Spatial Parity | $\Delta$ | Superconducting Properties |
|---|---|---|
| Even | $A^{J=0}_{1;j=1/2}$, $A^{J=0}_{1;j=3/2}$, $E^{J=2}_{j=3/2}$ | Coexisting superconducting order parameters. G.D. Time-reversal invariant states. |
| Odd | $T^{J=1}_{1;j=1/2}$, $T^{J=1,3}_{+;j=3/2}$, $T^{J=1,3}_{-;j=3/2}$, $T^{J=3}_{2;j=3/2}$ | Coexisting superconducting order parameters. Gapless point nodes along [110], [101], [011], [1$\bar{1}$0], [10$\bar{1}$], [01$\bar{1}$] axes. Time-reversal broken states. |
| Even | $E^{J=2}_{-;\frac{1}{2}\otimes\frac{3}{2}}$ | G.D. Time-reversal broken states. |
| Odd | $T^{J=2}_{2+;\frac{1}{2}\otimes\frac{3}{2}}$, $T^{J=1}_{1-;\frac{1}{2}\otimes\frac{3}{2}}$ | Coexisting superconducting order parameters. G.D. Time-reversal broken states. |

plies even in the case of multiple coexisting even-parity order parameters. The odd-parity order parameters, on the other hand, acquire gapless nodes in the quasiparticle dispersion, in a manner that respects the underlying cubic symmetry. We present a schematic of the realized gapless quasiparticle nodes for the order parameters in Fig. 4. The distinction in the location of the gapless nodes provides a means to distinguish the odd-parity order parameters. Indeed, the realized superconducting states arising from electrons belonging to either $j = \frac{3}{2}$ or $j = \frac{1}{2}$ sector are time-reversal invariant (broken) depending on if $J$ is even (odd).

For Cooper pairs formed from the combined $j = \frac{1}{2}, \frac{3}{2}$ sectors, gapped quasiparticle dispersions are realized for both even and odd parity order parameters. Unlike the order parameters where Cooper pairs are formed in the $j = \frac{1}{2}$ or $j = \frac{3}{2}$ sectors independently, odd-parity order parameters develop for even-$J$ Cooper pairs. This is an important distinction of this model from standard BCS-like instabilities formed from $j = \frac{1}{2}$ or from $j = \frac{3}{2}$ sectors. A further intriguing aspect from the combined model is that the odd and even $J$ Cooper pairs may coexist with each other. This is seen, for example, in the coexisting gaps formed from $T^{J=1}_{1-;\frac{1}{2}\otimes\frac{3}{2}}$ and $T^{J=2}_{2+;\frac{1}{2}\otimes\frac{3}{2}}$, where despite being odd under spatial parity, the quasiparticle spectrum is gapped on the Fermi surface. We contrast this with the nature of the gap functions obtained for odd-parity Cooper pairs, where distinct gapless nodes are along the various depicted cubic directions in Fig. 4. Indeed, both the realized superconducting states break time-reversal regardless of the even-ness of $J$, as is seen for both the $E^{J=2}_{-;\frac{1}{2}\otimes\frac{3}{2}}$ states, as well as for $T^{J=2}_{2+;\frac{1}{2}\otimes\frac{3}{2}}$, $T^{J=1}_{1-;\frac{1}{2}\otimes\frac{3}{2}}$ states. The origin of such unusual coexisting superconducting instabilities can be routed to the novel multipolar Kondo coupling that permitted the mixing of multi-orbital conduction electrons.

# 7 Conclusion

In this work, we examined the nature of superconducting instabilities originating from two-channel and novel multipolar Kondo interactions between multi-orbital conduction electrons and localized non-Kramers moments. Due to the multipolar nature of the localized moments, the spin and orbital of conduction electrons became intertwined, leading to pairing instabilities

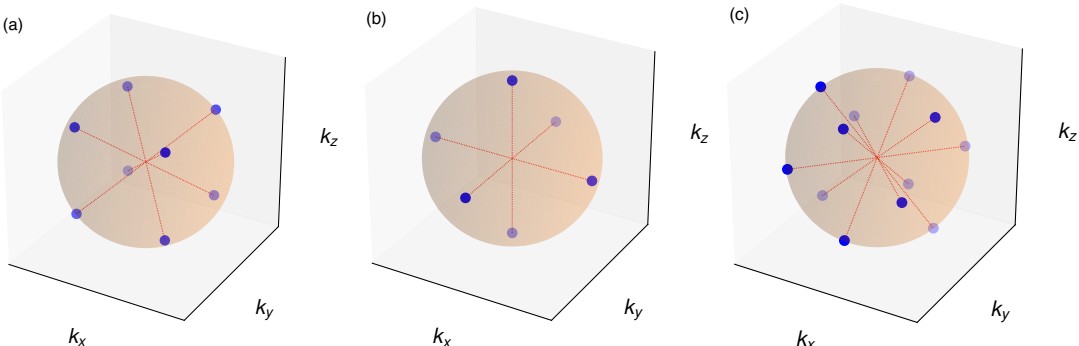

Figure 4: Gapless nodes in the Bogoliubov quasiparticle dispersions for odd-parity order parameters formed by fermions belonging to $j = 1/2$ or $j = 3/2$ sectors independently. The point nodes are indicated by the blue dots, while the orange sphere is the itinerant electron Fermi surface. The red-dashed lines are for ease of viewing the gapless point nodes along the various cubic axes. (a): Gapless nodes for $T_{+(1,2,3)}^{J=1,3}$, $T_2^{J=3}$ order parameters resulting from two-channel Kondo interaction are located along body diagonal [111] directions. (b): Gapless nodes for $T_{-(1,2,3)}^{J=1,3}$ order parameters resulting from two-channel Kondo interaction are located along primary cubic $\hat{e}_i$ axes. (c): Gapless nodes for $T_{+;j=3/2}^{J=1,3}$, $T_{-;j=3/2}^{J=1,3}$, $T_{2;j=3/2}^{J=3}$ resulting from novel Kondo interaction are located along the $[1 \pm 10]$, $[10 \pm 1]$, $[01 \pm 1]$ axes.

between effective $j = \frac{1}{2}, \frac{3}{2}$ the conduction electrons. Using group theoretic symmetry analysis, we characterized the variety of higher-angular momentum Cooper pairs according to the irreducible representations of the $O_h$ point group. The Cooper pairs arising from two-channel Kondo interactions are composed of electrons from the $j = \frac{1}{2}$, $j = \frac{3}{2}$ sectors independently, and possess even and odd spatial parity gap functions that follow from the even and odd-ness of their corresponding total angular momentum $J$. Indeed, the odd-parity quasiparticles possess point-nodal structures along various cubic directions in their dispersion. Intriguingly, Cooper pairs arising from the novel Kondo interaction leads to coexisting even-$J$ and odd-$J$ instabilities, which have even and odd spatial parity regardless of the even/odd-ness of the Cooper pair total angular momentum.

Our studies are broadly applicable to the rare-earth family of $Pr(Ti,V)_2Al_{20}$, $Pr(Ir,Rh)_2Zn_{20}$, where multipolar moments are situated on diamond lattice sites with well-localized Fermi surface formed by Al atoms. Indeed, our work may be employed as a theoretical guide to classify the superconducting instabilities occurring in the paramagnetic phase, which may be achieved via the application of hydrostatic pressure. Future directions of study would be to examine and classify the nature of the superconductivity coexisting within a multipolar ordered phase. Due to the reduced symmetry of the ordered phase, the above cubic irreps become reducible and it is natural to expect further variety of the pairing functions. Such studies would be highly relevant to observed superconductivity coexisting with quadrupolar ordered phase in $PrTi_2Al_{20}$ [64]. Indeed, microscopic details of the conduction electrons (such as the conduction electron band structure and Fermi surface in the paramagnetic phase) would be required to make direct connections with such coexistence experiments [75]. Understanding the topological nature of the superconducting states in both the paramagnetic and multipolar ordered phases may ultimately require such microscopic information, and would be an important future study.

## Acknowledgements

We thank Nazim Boudjada for illuminating discussions.

**Funding information** This work was supported by NSERC of Canada and the Centre for Quantum Materials.

## A Symmetry transformations of multipolar moments

The phenomenological order parameters transform as their microscopic counterparts under the generating elements of the $T_d$ point group (improper rotation $S_{4z}$ and a $C_3$ rotation along the [111] axis), namely [43, 53]:

$$\phi_x(\mathbf{r}) \xrightarrow{S_{4z}} -\phi_x(R_{S_{4z}}^{-1}\mathbf{r}), \tag{31}$$

$$\phi_y(\mathbf{r}) \xrightarrow{S_{4z}} \phi_y(R_{S_{4z}}^{-1}\mathbf{r}), \tag{32}$$

$$\phi_z(\mathbf{r}) \xrightarrow{S_{4z}} -\phi_z(R_{S_{4z}}^{-1}\mathbf{r}), \tag{33}$$

$$\phi_x(\mathbf{r}) \xrightarrow{C_{31}} -\frac{1}{2}\phi_x(R_{C_{31}}^{-1}\mathbf{r}) + \frac{\sqrt{3}}{2}\phi_y(R_{C_{31}}^{-1}\mathbf{r}), \tag{34}$$

$$\phi_y(\mathbf{r}) \xrightarrow{C_{31}} -\frac{\sqrt{3}}{2}\phi_x(R_{C_{31}}^{-1}\mathbf{r}) - \frac{1}{2}\phi_y(R_{C_{31}}^{-1}\mathbf{r}), \tag{35}$$

$$\phi_z(\mathbf{r}) \xrightarrow{C_{31}} \phi_z(R_{C_{31}}^{-1}\mathbf{r}). \tag{36}$$

The Fourier transform of these order parameters are given by,

$$\phi_{x,y,z}(\mathbf{q}) = \frac{1}{N}\sum_{\mathbf{r}} e^{-i\mathbf{q}\cdot\mathbf{r}}\phi_{x,y,z}(\mathbf{r}). \tag{37}$$

## B Multipolar Kondo interaction

The single-impurity model studied in Ref. [44] can be extended to a generalized coarse-grained lattice model where conduction band electrons uniformly couple to the ferro-multipolar order parameters in a manner respecting the local $T_d$ symmetry of the moments,

$$H_K = 2\sum_{\mathbf{r}} \sum_{d=x,y,z} c_{\mathbf{r}\mu}^{\dagger} \Gamma_{\mu\nu}^d c_{\mathbf{r}\nu} \phi_d(\mathbf{r}), \tag{38}$$

where

$$\begin{aligned}
\Gamma^x &= J_1(\lambda^1 + \lambda^{27}) - J_2(\lambda^4 - \lambda^{30}), \\
\Gamma^y &= J_1(\lambda^3 + \lambda^{29}) - J_2(\lambda^6 - \lambda^{32}), \\
\Gamma^z &= -J_3(\lambda^2 + \lambda^{28}).
\end{aligned} \tag{39}$$

Here, $J_1, J_2, J_3$ are the Kondo couplings to the multipolar moments, $\mathbf{r}$ denotes the coarse-grained spatial coordinate, and $\lambda^W$ are the SU(6) Gell-Mann generators $W = \{1, 2..., 35\}$ with normalization such that $\text{tr}[\lambda^a\lambda^b] = \frac{1}{2}\delta^{ab}$. In the single impurity limit, the IR fixed points are described by (i) $J_1 = J_3 \neq 0; J_2 = 0$ (two-channel Kondo interaction fixed point), and

(ii) $J_2 = J_3 \neq 0; J_3 = 0$ (novel fixed point). In terms of the coefficients studied in Ref. [44]: $J_1 \equiv -\frac{1}{\sqrt{3}}K_1 + 2K_2$, $J_2 \equiv \sqrt{\frac{2}{3}}K_1 + \sqrt{2}K_2$, and $J_3 \equiv \sqrt{3}K_3$. We note that the factor of 2 in Eq. 38 is cancelled out by the factor of $\frac{1}{2}$ introduced in Eq. 5 to include both $\phi_{x,y,z}(\mathbf{q})$ and $\phi_{x,y,z}(-\mathbf{q})$ coupling to the fermionic bilinears. We note that the 1/2-factor normalization of the SU(6) Gell-Mann generators is absorbed into the definition of $J_{1,2,3}$ henceforth for simplicity. The conduction electron basis employed in Eq. 38 is,

$$\vec{c}_\mathbf{r}^\top = \left( c_{\mathbf{r};\frac{3}{2},\frac{-3}{2}} \ c_{\mathbf{r};\frac{3}{2},\frac{1}{2}} \ c_{\mathbf{r};\frac{1}{2},\frac{1}{2}} \ c_{\mathbf{r};\frac{3}{2},\frac{3}{2}} \ c_{\mathbf{r};\frac{3}{2},\frac{-1}{2}} \ c_{\mathbf{r};\frac{1}{2},\frac{-1}{2}} \right), \tag{40}$$

where the subscript $\mathbf{r}; j, j_z$ for the fermionic operator indicates the coarse-grained spatial coordinate, total-angular momentum $j$, and $z$-component projection of the total angular momentum $j_z$, respectively. The Fourier transform of the conduction electron fields is given by,

$$c_{\mathbf{r}\mu} = \frac{1}{\sqrt{N}} \sum_\mathbf{k} e^{i\mathbf{k}\cdot\mathbf{r}} c_{\mathbf{k}\mu}. \tag{41}$$

## C  Effective electron-electron interaction from multipolar Kondo interaction

The total path integral (composed of conduction electron action, Kondo coupling, and multipolar fluctiations) is given by,

$$Z = \int \mathcal{D}[\overline{c}, c]\mathcal{D}[\overline{\phi}, \phi] e^{-(S_c + S_0 + S_K)}$$

$$= \int \mathcal{D}[\overline{c}, c] e^{-S_c} \mathcal{D}[\overline{\phi}, \phi] e^{-\int_{\tau,\mathbf{q}} \left[ \sum_{\mu\nu} \phi_\mu(-\mathbf{q})M_{\mu\nu}\phi_\nu(\mathbf{q}) + \sum_\mu \phi_\mu(\mathbf{q})\Gamma^\mu(\mathbf{q}) + \sum_\mu \phi_\mu(-\mathbf{q})\Gamma^\mu(-\mathbf{q}) \right]}, \tag{42}$$

where

$$\vec{\Gamma}(\mathbf{q}) \equiv \begin{bmatrix} \sum_{\mathbf{k};\alpha,\beta} \overline{c}_{\mathbf{k}+\mathbf{q},\alpha} \Gamma^x_{\alpha\beta} c_{\mathbf{k},\beta} \\ \sum_{\mathbf{k};\alpha,\beta} \overline{c}_{\mathbf{k}+\mathbf{q},\alpha} \Gamma^y_{\alpha\beta} c_{\mathbf{k},\beta} \\ \sum_{\mathbf{k};\alpha,\beta} \overline{c}_{\mathbf{k}+\mathbf{q},\alpha} \Gamma^z_{\alpha\beta} c_{\mathbf{k},\beta} \end{bmatrix}, \tag{43}$$

and the measure is given by

$$\mathcal{D}[\overline{\phi}, \phi] = \prod_{\mu=\{x,y,z\}} \mathcal{D}[\overline{\phi}_\mu, \phi_\mu]$$

$$= \prod_{\mu=\{x,y,z\}} \left( \lim_{N\to\infty} \prod_{l=1}^N \frac{d\overline{\phi}_{\mu,l} d\phi_{\mu,l}}{2\pi i} \right)$$

$$\equiv \prod_{\mu=\{x,y,z\}} \frac{d\overline{\phi}_\mu d\phi_\mu}{2\pi i}. \tag{44}$$

Integrating out the bosonic field variable using the identity:

$$\int \prod_\alpha \frac{d\overline{b}_\alpha db_\alpha}{2\pi i} e^{-(\overline{b}_\alpha M_{\alpha\beta} b_\beta - \overline{j}_\alpha b_\alpha - \overline{b}_\alpha j_\alpha)} = \frac{e^{\overline{j}\cdot M^{-1} j}}{\det[M]}, \tag{45}$$

we thus arrive at $Z = \int \mathcal{D}[\bar{c}, c] e^{-S_c} e^{-S_{\text{eff}}}$ with the effective interaction

$$-S_{\text{eff}} = \int_{\tau, \mathbf{k}, \mathbf{k}', \mathbf{q}} (\mathcal{V}_{\alpha\beta\gamma\delta})_{\mathbf{q}} c^{\dagger}_{\mathbf{k}+\mathbf{q}, \alpha} c_{\mathbf{k}, \beta} c^{\dagger}_{\mathbf{k}'-\mathbf{q}, \gamma} c_{\mathbf{k}', \delta} , \tag{46}$$

where the interaction vertex is,

$$(\mathcal{V}_{\alpha\beta\gamma\delta})_{\mathbf{q}} = \Bigg[ f_0(\mathbf{q}) \Big( \Gamma^x_{\alpha\beta} \Gamma^x_{\gamma\delta} + \Gamma^y_{\alpha\beta} \Gamma^y_{\gamma\delta} \Big) + f_1(\mathbf{q}) \Big( \Gamma^z_{\alpha\beta} \Gamma^z_{\gamma\delta} \Big) - f_{2\nu}(\mathbf{q}) \Big( \Gamma^x_{\alpha\beta} \Gamma^x_{\gamma\delta} - \Gamma^y_{\alpha\beta} \Gamma^y_{\gamma\delta} \Big)$$

$$- f_{2\mu}(\mathbf{q}) \Big( \Gamma^x_{\alpha\beta} \Gamma^y_{\gamma\delta} + \Gamma^y_{\alpha\beta} \Gamma^x_{\gamma\delta} \Big) \Bigg], \tag{47}$$

and the interaction potential terms are,

$$f_0(\mathbf{q}) \equiv \frac{\left( a_0 \mathbf{q}^2 + m_{\mathcal{Q}} \right)}{\left( a_0 \mathbf{q}^2 + m_{\mathcal{Q}} \right)^2 - a_2^2 \left( q_\mu^4 + q_\nu^4 \right)} , \tag{48}$$

$$f_1(\mathbf{q}) \equiv \frac{1}{a_1(\mathbf{q})^2 + m_{\mathcal{O}}} , \tag{49}$$

$$f_{2\mu}(\mathbf{q}) \equiv \frac{a_2 q_\mu^2}{\left( a_0 \mathbf{q}^2 + m_{\mathcal{Q}} \right)^2 - a_2^2 \left( q_\mu^4 + q_\nu^4 \right)} , \tag{50}$$

$$f_{2\nu}(\mathbf{q}) \equiv \frac{a_2 q_\nu^2}{\left( a_0 \mathbf{q}^2 + m_{\mathcal{Q}} \right)^2 - a_2^2 \left( q_\mu^4 + q_\nu^4 \right)} . \tag{51}$$

We note that once again, $q_\nu^2 \equiv \frac{1}{2}(2q_z^2 - q_x^2 - q_y^2)$ and $q_\mu^2 \equiv \frac{\sqrt{3}}{2}(q_x^2 - q_y^2)$. This interaction is prepared for investigating superconducting instabilities by (as described in the main text) (i) normal ordering the interaction $: c^{\dagger}_{\mathbf{k}+\mathbf{q}, \alpha} c_{\mathbf{k}, \beta} c^{\dagger}_{\mathbf{k}'-\mathbf{q}, \gamma} c_{\mathbf{k}', \delta} : \; = \; c^{\dagger}_{\mathbf{k}+\mathbf{q}, \alpha} c^{\dagger}_{\mathbf{k}'-\mathbf{q}, \gamma} c_{\mathbf{k}', \delta} c_{\mathbf{k}, \beta}$, and (ii) projecting the Cooper pairs to being formed by opposite momentum electrons (in the same spirit as BCS theory), $\mathbf{k}' = -\mathbf{k}$. This leads to the effective interaction Hamiltonian in Eq. 7.

## D  Symmetry decomposition of total angular momentum Cooper pair states

The total angular momentum Cooper pair states can be elegantly decoupled into the irreps of the point group $O_h$. The $O_h$ point group contains 48 elements $O_h = \{T_d, \mathbb{I} \times T_d\}$, where $\mathbb{I}$ is inversion, and the $T_d$ elements are, $T_d = \{E, C^+_{31}, C^-_{31}, C^+_{32}, C^-_{32}, C^+_{33}, C^-_{33}, C^+_{34}, C^-_{34}, C_{2x}, C_{2y}, C_{2z}, S^+_{4x}, S^+_{4y}, S^+_{4z}, S^-_{4x}, S^-_{4y}, S^-_{4z}, \sigma_{da}, \sigma_{db}, \sigma_{dc}, \sigma_{dd}, \sigma_{de}, \sigma_{df}\}$ where we use the standard Schoenflies notation to denote the symmetry elements.

The total angular momentum states can be used to construct basis functions for each of the irreps using the projection operator,

$$P^i_\alpha = \frac{d_\alpha}{g} \sum_G \langle \Gamma^i_\alpha | G | \Gamma^i_\alpha \rangle^* , \tag{52}$$

where $i$ labels the basis function of the $d_\alpha$-dimensional irrep $\Gamma_\alpha$ (i.e. the basis function is $|\Gamma^i_\alpha\rangle$). $g = 48$ is the order of the point group of interest, and $\alpha$ runs over the possible irreps in $O_h$. In order to employ the projection operator, the matrix elements $\langle \Gamma^i_\alpha | G | \Gamma^i_\alpha \rangle^*$ in Eq. 52 need to be extracted. Since the basis functions of the irreps in terms of cartesian basis states is known (for example, for $T_{2g}$, the basis functions are $|\Gamma_{T_{2g}}\rangle = \{yz, xz, xy\}$), they are employed (along with the cartesian representation of the elements of $O_h$) to compute the aforementioned matrix element, and thus the projection operator.

## E  Cooper pair composition from $j = \frac{1}{2}, \frac{3}{2}$ sectors of conduction electrons

The Cooper pair operator associated with a particular irrep can be constructed from different means depending on the angular momentum $j$ associated with the individual conduction electrons. We list the possibilities below, and use the notation of

$$\Delta_{\mathbf{k}}^{\dagger}\left(j_1, m_{j_1} \Big| j_2, m_{j_2}\right) = c_{j_1, m_{j_1}; \mathbf{k}}^{\dagger} c_{j_2, m_{j_2}; -\mathbf{k}}^{\dagger}, \tag{53}$$

for brevity. We organize the Cooper pairs in terms of even or odd total angular momentum $J$, as well as the conduction electrons sector from which they are constructed from.

Even J: $\frac{1}{2} \otimes \frac{1}{2}$

$$(A_1^{\dagger; J=0})_{\mathbf{k}} = \frac{1}{2}\left[\Delta_{\mathbf{k}}^{\dagger}\left(\frac{1}{2}, \frac{1}{2}\Big|\frac{1}{2}, \frac{-1}{2}\right) - \Delta_{\mathbf{k}}^{\dagger}\left(\frac{1}{2}, \frac{-1}{2}\Big|\frac{1}{2}, \frac{1}{2}\right)\right]. \tag{54}$$

Odd J: $\frac{1}{2} \otimes \frac{1}{2}$

$$(T_{1(1)}^{\dagger; J=1})_{\mathbf{k}} = \frac{1}{\sqrt{2}}\left[\Delta_{\mathbf{k}}^{\dagger}\left(\frac{1}{2}, \frac{1}{2}\Big|\frac{1}{2}, \frac{1}{2}\right) - \Delta_{\mathbf{k}}^{\dagger}\left(\frac{1}{2}, \frac{-1}{2}\Big|\frac{1}{2}, \frac{-1}{2}\right)\right], \tag{55}$$

$$(T_{1(2)}^{\dagger; J=1})_{\mathbf{k}} = \frac{i}{\sqrt{2}}\left[\Delta_{\mathbf{k}}^{\dagger}\left(\frac{1}{2}, \frac{1}{2}\Big|\frac{1}{2}, \frac{1}{2}\right) + \Delta_{\mathbf{k}}^{\dagger}\left(\frac{1}{2}, \frac{-1}{2}\Big|\frac{1}{2}, \frac{-1}{2}\right)\right], \tag{56}$$

$$(T_{1(3)}^{\dagger; J=1})_{\mathbf{k}} = \frac{1}{\sqrt{2}}\left[\Delta_{\mathbf{k}}^{\dagger}\left(\frac{1}{2}, \frac{1}{2}\Big|\frac{1}{2}, \frac{-1}{2}\right) + \Delta_{\mathbf{k}}^{\dagger}\left(\frac{1}{2}, \frac{-1}{2}\Big|\frac{1}{2}, \frac{1}{2}\right)\right]. \tag{57}$$

Even J: $\frac{3}{2} \otimes \frac{3}{2}$

$$\begin{aligned}
(A_1^{\dagger; J=0})_{\mathbf{k}} = \frac{1}{2}\Bigg[&\Delta_{\mathbf{k}}^{\dagger}\left(\frac{3}{2}, \frac{3}{2}\Big|\frac{3}{2}, \frac{-3}{2}\right) - \Delta_{\mathbf{k}}^{\dagger}\left(\frac{3}{2}, \frac{1}{2}\Big|\frac{3}{2}, \frac{-1}{2}\right) \\
&+ \Delta_{\mathbf{k}}^{\dagger}\left(\frac{3}{2}, \frac{-1}{2}\Big|\frac{3}{2}, \frac{1}{2}\right) - \Delta_{\mathbf{k}}^{\dagger}\left(\frac{3}{2}, \frac{-3}{2}\Big|\frac{3}{2}, \frac{3}{2}\right)\Bigg],
\end{aligned} \tag{58}$$

$$\begin{aligned}
(E_{(1)}^{\dagger; J=2})_{\mathbf{k}} = \frac{1}{2}\Bigg[&\Delta_{\mathbf{k}}^{\dagger}\left(\frac{3}{2}, \frac{3}{2}\Big|\frac{3}{2}, \frac{1}{2}\right) - \Delta_{\mathbf{k}}^{\dagger}\left(\frac{3}{2}, \frac{1}{2}\Big|\frac{3}{2}, \frac{3}{2}\right) \\
&+ \Delta_{\mathbf{k}}^{\dagger}\left(\frac{3}{2}, \frac{-1}{2}\Big|\frac{3}{2}, \frac{-3}{2}\right) - \Delta_{\mathbf{k}}^{\dagger}\left(\frac{3}{2}, \frac{-3}{2}\Big|\frac{3}{2}, \frac{-1}{2}\right)\Bigg],
\end{aligned} \tag{59}$$

$$\begin{aligned}
(E_{(2)}^{\dagger; J=2})_{\mathbf{k}} = \frac{1}{2}\Bigg[&\Delta_{\mathbf{k}}^{\dagger}\left(\frac{3}{2}, \frac{3}{2}\Big|\frac{3}{2}, \frac{-3}{2}\right) + \Delta_{\mathbf{k}}^{\dagger}\left(\frac{3}{2}, \frac{1}{2}\Big|\frac{3}{2}, \frac{-1}{2}\right) \\
&- \Delta_{\mathbf{k}}^{\dagger}\left(\frac{3}{2}, \frac{-1}{2}\Big|\frac{3}{2}, \frac{1}{2}\right) - \Delta_{\mathbf{k}}^{\dagger}\left(\frac{3}{2}, \frac{-3}{2}\Big|\frac{3}{2}, \frac{3}{2}\right)\Bigg],
\end{aligned} \tag{60}$$

$$\begin{aligned}
(T_{2(1)}^{\dagger; J=2})_{\mathbf{k}} = \frac{i}{2}\Bigg[&\Delta_{\mathbf{k}}^{\dagger}\left(\frac{3}{2}, \frac{3}{2}\Big|\frac{3}{2}, \frac{-1}{2}\right) - \Delta_{\mathbf{k}}^{\dagger}\left(\frac{3}{2}, \frac{-1}{2}\Big|\frac{3}{2}, \frac{3}{2}\right) \\
&+ \Delta_{\mathbf{k}}^{\dagger}\left(\frac{3}{2}, \frac{1}{2}\Big|\frac{3}{2}, \frac{-3}{2}\right) - \Delta_{\mathbf{k}}^{\dagger}\left(\frac{3}{2}, \frac{-3}{2}\Big|\frac{3}{2}, \frac{1}{2}\right)\Bigg],
\end{aligned} \tag{61}$$

$$(T_{2(2)}^{\dagger;J=2})_{\mathbf{k}} = \frac{1}{2}\left[\Delta_{\mathbf{k}}^{\dagger}\left(\frac{3}{2},\frac{3}{2}\bigg|\frac{3}{2},\frac{-1}{2}\right) - \Delta_{\mathbf{k}}^{\dagger}\left(\frac{3}{2},\frac{-1}{2}\bigg|\frac{3}{2},\frac{3}{2}\right)\right.$$

$$\left. - \Delta_{\mathbf{k}}^{\dagger}\left(\frac{3}{2},\frac{1}{2}\bigg|\frac{3}{2},\frac{-3}{2}\right) + \Delta_{\mathbf{k}}^{\dagger}\left(\frac{3}{2},\frac{-3}{2}\bigg|\frac{3}{2},\frac{1}{2}\right)\right], \tag{62}$$

$$(T_{2(3)}^{\dagger;J=2})_{\mathbf{k}} = \frac{i}{2}\left[\Delta_{\mathbf{k}}^{\dagger}\left(\frac{3}{2},\frac{3}{2}\bigg|\frac{3}{2},\frac{1}{2}\right) - \Delta_{\mathbf{k}}^{\dagger}\left(\frac{3}{2},\frac{1}{2}\bigg|\frac{3}{2},\frac{3}{2}\right)\right.$$

$$\left. - \Delta_{\mathbf{k}}^{\dagger}\left(\frac{3}{2},\frac{-1}{2}\bigg|\frac{3}{2},\frac{-3}{2}\right) + \Delta_{\mathbf{k}}^{\dagger}\left(\frac{3}{2},\frac{-3}{2}\bigg|\frac{3}{2},\frac{-1}{2}\right)\right]. \tag{63}$$

Odd J: $\frac{3}{2} \otimes \frac{3}{2}$

$$(T_{1(1)}^{\dagger;J=1})_{\mathbf{k}} = \frac{1}{2}\sqrt{\frac{3}{5}}\left[\Delta_{\mathbf{k}}^{\dagger}\left(\frac{3}{2},\frac{3}{2}\bigg|\frac{3}{2},\frac{-1}{2}\right) + \Delta_{\mathbf{k}}^{\dagger}\left(\frac{3}{2},\frac{-1}{2}\bigg|\frac{3}{2},\frac{3}{2}\right)\right.$$

$$\left. - \Delta_{\mathbf{k}}^{\dagger}\left(\frac{3}{2},\frac{1}{2}\bigg|\frac{3}{2},\frac{-3}{2}\right) - \Delta_{\mathbf{k}}^{\dagger}\left(\frac{3}{2},\frac{-3}{2}\bigg|\frac{3}{2},\frac{1}{2}\right)\right]$$

$$- \frac{1}{\sqrt{5}}\left[\Delta_{\mathbf{k}}^{\dagger}\left(\frac{3}{2},\frac{1}{2}\bigg|\frac{3}{2},\frac{1}{2}\right) - \Delta_{\mathbf{k}}^{\dagger}\left(\frac{3}{2},\frac{-1}{2}\bigg|\frac{3}{2},\frac{-1}{2}\right)\right], \tag{64}$$

$$(T_{1(2)}^{\dagger;J=1})_{\mathbf{k}} = \frac{i}{2}\sqrt{\frac{3}{5}}\left[\Delta_{\mathbf{k}}^{\dagger}\left(\frac{3}{2},\frac{3}{2}\bigg|\frac{3}{2},\frac{-1}{2}\right) + \Delta_{\mathbf{k}}^{\dagger}\left(\frac{3}{2},\frac{-1}{2}\bigg|\frac{3}{2},\frac{3}{2}\right)\right.$$

$$\left. + \Delta_{\mathbf{k}}^{\dagger}\left(\frac{3}{2},\frac{1}{2}\bigg|\frac{3}{2},\frac{-3}{2}\right) + \Delta_{\mathbf{k}}^{\dagger}\left(\frac{3}{2},\frac{-3}{2}\bigg|\frac{3}{2},\frac{1}{2}\right)\right]$$

$$- \frac{i}{\sqrt{5}}\left[\Delta_{\mathbf{k}}^{\dagger}\left(\frac{3}{2},\frac{1}{2}\bigg|\frac{3}{2},\frac{1}{2}\right) + \Delta_{\mathbf{k}}^{\dagger}\left(\frac{3}{2},\frac{-1}{2}\bigg|\frac{3}{2},\frac{-1}{2}\right)\right], \tag{65}$$

$$(T_{1(3)}^{\dagger;J=1})_{\mathbf{k}} = \frac{1}{2\sqrt{5}}\left[3\Delta_{\mathbf{k}}^{\dagger}\left(\frac{3}{2},\frac{3}{2}\bigg|\frac{3}{2},\frac{-3}{2}\right) - \Delta_{\mathbf{k}}^{\dagger}\left(\frac{3}{2},\frac{1}{2}\bigg|\frac{3}{2},\frac{-1}{2}\right)\right.$$

$$\left. - \Delta_{\mathbf{k}}^{\dagger}\left(\frac{3}{2},\frac{-1}{2}\bigg|\frac{3}{2},\frac{1}{2}\right) + 3\Delta_{\mathbf{k}}^{\dagger}\left(\frac{3}{2},\frac{-3}{2}\bigg|\frac{3}{2},\frac{3}{2}\right)\right], \tag{66}$$

$$(A_{2}^{\dagger;J=3})_{\mathbf{k}} = \frac{i}{2}\left[\Delta_{\mathbf{k}}^{\dagger}\left(\frac{3}{2},\frac{3}{2}\bigg|\frac{3}{2},\frac{1}{2}\right) + \Delta_{\mathbf{k}}^{\dagger}\left(\frac{3}{2},\frac{1}{2}\bigg|\frac{3}{2},\frac{3}{2}\right)\right.$$

$$\left. - \Delta_{\mathbf{k}}^{\dagger}\left(\frac{3}{2},\frac{-1}{2}\bigg|\frac{3}{2},\frac{-3}{2}\right) - \Delta_{\mathbf{k}}^{\dagger}\left(\frac{3}{2},\frac{-3}{2}\bigg|\frac{3}{2},\frac{-1}{2}\right)\right], \tag{67}$$

$$(T_{1(1)}^{\dagger;J=3})_{\mathbf{k}} = \frac{\sqrt{5}}{4}\left(\Delta_{\mathbf{k}}^{\dagger}\left(\frac{3}{2},\frac{3}{2}\bigg|\frac{3}{2},\frac{3}{2}\right) - \Delta_{\mathbf{k}}^{\dagger}\left(\frac{3}{2},\frac{-3}{2}\bigg|\frac{3}{2},\frac{-3}{2}\right)\right)$$

$$- \frac{3}{4\sqrt{5}}\left(\Delta_{\mathbf{k}}^{\dagger}\left(\frac{3}{2},\frac{1}{2}\bigg|\frac{3}{2},\frac{1}{2}\right) - \Delta_{\mathbf{k}}^{\dagger}\left(\frac{3}{2},\frac{-1}{2}\bigg|\frac{3}{2},\frac{-1}{2}\right)\right]$$

$$- \frac{1}{4}\sqrt{\frac{3}{5}}\left[\Delta_{\mathbf{k}}^{\dagger}\left(\frac{3}{2},\frac{3}{2}\bigg|\frac{3}{2},\frac{-1}{2}\right) + \Delta_{\mathbf{k}}^{\dagger}\left(\frac{3}{2},\frac{-1}{2}\bigg|\frac{3}{2},\frac{3}{2}\right)\right.$$

$$\left. - \Delta_{\mathbf{k}}^{\dagger}\left(\frac{3}{2},\frac{1}{2}\bigg|\frac{3}{2},\frac{-3}{2}\right) - \Delta_{\mathbf{k}}^{\dagger}\left(\frac{3}{2},\frac{-3}{2}\bigg|\frac{3}{2},\frac{1}{2}\right)\right], \tag{68}$$

$$(T_{1(2)}^{\dagger;J=3})_{\mathbf{k}} = i\frac{\sqrt{5}}{4}\left[\Delta_{\mathbf{k}}^{\dagger}\left(\frac{3}{2},\frac{3}{2}\Big|\frac{3}{2},\frac{3}{2}\right) + \Delta_{\mathbf{k}}^{\dagger}\left(\frac{3}{2},\frac{-3}{2}\Big|\frac{3}{2},\frac{-3}{2}\right)\right)$$

$$+ \frac{3}{4\sqrt{5}}\left(\Delta_{\mathbf{k}}^{\dagger}\left(\frac{3}{2},\frac{1}{2}\Big|\frac{3}{2},\frac{1}{2}\right) + \Delta_{\mathbf{k}}^{\dagger}\left(\frac{3}{2},\frac{-1}{2}\Big|\frac{3}{2},\frac{-1}{2}\right)\right)\right]$$

$$+ i\frac{1}{4}\sqrt{\frac{3}{5}}\left[\Delta_{\mathbf{k}}^{\dagger}\left(\frac{3}{2},\frac{3}{2}\Big|\frac{3}{2},\frac{-1}{2}\right) + \Delta_{\mathbf{k}}^{\dagger}\left(\frac{3}{2},\frac{-1}{2}\Big|\frac{3}{2},\frac{3}{2}\right)\right.$$

$$\left. + \Delta_{\mathbf{k}}^{\dagger}\left(\frac{3}{2},\frac{1}{2}\Big|\frac{3}{2},\frac{-3}{2}\right) + \Delta_{\mathbf{k}}^{\dagger}\left(\frac{3}{2},\frac{-3}{2}\Big|\frac{3}{2},\frac{1}{2}\right)\right], \tag{69}$$

$$(T_{1(3)}^{\dagger;J=3})_{\mathbf{k}} = \frac{1}{2\sqrt{5}}\left[\Delta_{\mathbf{k}}^{\dagger}\left(\frac{3}{2},\frac{3}{2}\Big|\frac{3}{2},\frac{-3}{2}\right) + 3\Delta_{\mathbf{k}}^{\dagger}\left(\frac{3}{2},\frac{1}{2}\Big|\frac{3}{2},\frac{-1}{2}\right)\right.$$

$$\left. + 3\Delta_{\mathbf{k}}^{\dagger}\left(\frac{3}{2},\frac{-1}{2}\Big|\frac{3}{2},\frac{1}{2}\right) + \Delta_{\mathbf{k}}^{\dagger}\left(\frac{3}{2},\frac{-3}{2}\Big|\frac{3}{2},\frac{3}{2}\right)\right], \tag{70}$$

$$(T_{2(1)}^{\dagger;J=3})_{\mathbf{k}} = \frac{\sqrt{3}}{4}\left[\Delta_{\mathbf{k}}^{\dagger}\left(\frac{3}{2},\frac{3}{2}\Big|\frac{3}{2},\frac{3}{2}\right) + \Delta_{\mathbf{k}}^{\dagger}\left(\frac{3}{2},\frac{1}{2}\Big|\frac{3}{2},\frac{1}{2}\right)\right.$$

$$\left. - \Delta_{\mathbf{k}}^{\dagger}\left(\frac{3}{2},\frac{-1}{2}\Big|\frac{3}{2},\frac{-1}{2}\right) - \Delta_{\mathbf{k}}^{\dagger}\left(\frac{3}{2},\frac{-3}{2}\Big|\frac{3}{2},\frac{-3}{2}\right)\right]$$

$$+ \frac{1}{4}\left[\Delta_{\mathbf{k}}^{\dagger}\left(\frac{3}{2},\frac{3}{2}\Big|\frac{3}{2},\frac{-1}{2}\right) + \Delta_{\mathbf{k}}^{\dagger}\left(\frac{3}{2},\frac{-1}{2}\Big|\frac{3}{2},\frac{3}{2}\right)\right.$$

$$\left. - \Delta_{\mathbf{k}}^{\dagger}\left(\frac{3}{2},\frac{1}{2}\Big|\frac{3}{2},\frac{-3}{2}\right) - \Delta_{\mathbf{k}}^{\dagger}\left(\frac{3}{2},\frac{-3}{2}\Big|\frac{3}{2},\frac{1}{2}\right)\right], \tag{71}$$

$$(T_{2(2)}^{\dagger;J=3})_{\mathbf{k}} = i\frac{\sqrt{3}}{4}\left[\Delta_{\mathbf{k}}^{\dagger}\left(\frac{3}{2},\frac{3}{2}\Big|\frac{3}{2},\frac{3}{2}\right) - \Delta_{\mathbf{k}}^{\dagger}\left(\frac{3}{2},\frac{1}{2}\Big|\frac{3}{2},\frac{1}{2}\right)\right.$$

$$\left. - \Delta_{\mathbf{k}}^{\dagger}\left(\frac{3}{2},\frac{-1}{2}\Big|\frac{3}{2},\frac{-1}{2}\right) + \Delta_{\mathbf{k}}^{\dagger}\left(\frac{3}{2},\frac{-3}{2}\Big|\frac{3}{2},\frac{-3}{2}\right)\right]$$

$$- i\frac{1}{4}\left[\Delta_{\mathbf{k}}^{\dagger}\left(\frac{3}{2},\frac{3}{2}\Big|\frac{3}{2},\frac{-1}{2}\right) + \Delta_{\mathbf{k}}^{\dagger}\left(\frac{3}{2},\frac{-1}{2}\Big|\frac{3}{2},\frac{3}{2}\right)\right.$$

$$\left. + \Delta_{\mathbf{k}}^{\dagger}\left(\frac{3}{2},\frac{1}{2}\Big|\frac{3}{2},\frac{-3}{2}\right) + \Delta_{\mathbf{k}}^{\dagger}\left(\frac{3}{2},\frac{-3}{2}\Big|\frac{3}{2},\frac{1}{2}\right)\right], \tag{72}$$

$$(T_{2(3)}^{\dagger;J=3})_{\mathbf{k}} = \frac{\sqrt{3}}{4}\left[\Delta_{\mathbf{k}}^{\dagger}\left(\frac{3}{2},\frac{3}{2}\Big|\frac{3}{2},\frac{1}{2}\right) + \Delta_{\mathbf{k}}^{\dagger}\left(\frac{3}{2},\frac{1}{2}\Big|\frac{3}{2},\frac{3}{2}\right)\right.$$

$$\left. - \Delta_{\mathbf{k}}^{\dagger}\left(\frac{3}{2},\frac{-1}{2}\Big|\frac{3}{2},\frac{-3}{2}\right) - \Delta_{\mathbf{k}}^{\dagger}\left(\frac{3}{2},\frac{-3}{2}\Big|\frac{3}{2},\frac{-1}{2}\right)\right]. \tag{73}$$

For Cooper pairs created from the mixed sector of $j = 1/2$ and $j = 3/2$ the ordering of the fermionic operators are important. We use the $\tilde{\mathcal{M}}$ notation to indicate operators where the order of the operators is interchanged. Even J: $\frac{3}{2} \otimes \frac{1}{2}$; $[\frac{1}{2} \otimes \frac{3}{2}]$

$$(E_{(1)}^{\dagger;J=2})_{\mathbf{k}} = \frac{i}{\sqrt{2}}\left[\Delta_{\mathbf{k}}^{\dagger}\left(\frac{3}{2},\frac{3}{2}\Big|\frac{1}{2},\frac{1}{2}\right) + \Delta_{\mathbf{k}}^{\dagger}\left(\frac{3}{2},\frac{-3}{2}\Big|\frac{1}{2},\frac{-1}{2}\right)\right], \tag{74}$$

$$(E_{(2)}^{\dagger;J=2})_{\mathbf{k}} = \frac{i}{\sqrt{2}}\left[\Delta_{\mathbf{k}}^{\dagger}\left(\frac{3}{2},\frac{1}{2}\Big|\frac{1}{2},\frac{-1}{2}\right) + \Delta_{\mathbf{k}}^{\dagger}\left(\frac{3}{2},\frac{-1}{2}\Big|\frac{1}{2},\frac{1}{2}\right)\right], \tag{75}$$

$$(\tilde{E}_{(1)}^{\dagger;J=2})_{\mathbf{k}} = \frac{i}{\sqrt{2}}\left[\Delta_{\mathbf{k}}^{\dagger}\left(\frac{1}{2},\frac{1}{2}\middle|\frac{3}{2},\frac{3}{2}\right) + \Delta_{\mathbf{k}}^{\dagger}\left(\frac{1}{2},\frac{-1}{2}\middle|\frac{3}{2},\frac{-3}{2}\right)\right], \tag{76}$$

$$(\tilde{E}_{(2)}^{\dagger;J=2})_{\mathbf{k}} = \frac{i}{\sqrt{2}}\left[\Delta_{\mathbf{k}}^{\dagger}\left(\frac{1}{2},\frac{-1}{2}\middle|\frac{3}{2},\frac{1}{2}\right) + \Delta_{\mathbf{k}}^{\dagger}\left(\frac{1}{2},\frac{1}{2}\middle|\frac{3}{2},\frac{-1}{2}\right)\right], \tag{77}$$

$$(T_{2(1)}^{\dagger;J=2})_{\mathbf{k}} = \frac{1}{2\sqrt{2}}\left[\Delta_{\mathbf{k}}^{\dagger}\left(\frac{3}{2},\frac{3}{2}\middle|\frac{1}{2},\frac{-1}{2}\right) + \Delta_{\mathbf{k}}^{\dagger}\left(\frac{3}{2},\frac{-3}{2}\middle|\frac{1}{2},\frac{1}{2}\right)\right.$$
$$\left. + \sqrt{3}\left(\Delta_{\mathbf{k}}^{\dagger}\left(\frac{3}{2},\frac{1}{2}\middle|\frac{1}{2},\frac{1}{2}\right) + \Delta_{\mathbf{k}}^{\dagger}\left(\frac{3}{2},\frac{-1}{2}\middle|\frac{1}{2},\frac{-1}{2}\right)\right)\right], \tag{78}$$

$$(T_{2(2)}^{\dagger;J=2})_{\mathbf{k}} = \frac{i}{2\sqrt{2}}\left[\Delta_{\mathbf{k}}^{\dagger}\left(\frac{3}{2},\frac{3}{2}\middle|\frac{1}{2},\frac{-1}{2}\right) - \Delta_{\mathbf{k}}^{\dagger}\left(\frac{3}{2},\frac{-3}{2}\middle|\frac{1}{2},\frac{1}{2}\right)\right.$$
$$\left. + \sqrt{3}\left(\Delta_{\mathbf{k}}^{\dagger}\left(\frac{3}{2},\frac{1}{2}\middle|\frac{1}{2},\frac{1}{2}\right) - \Delta_{\mathbf{k}}^{\dagger}\left(\frac{3}{2},\frac{-1}{2}\middle|\frac{1}{2},\frac{-1}{2}\right)\right)\right], \tag{79}$$

$$(T_{2(3)}^{\dagger;J=2})_{\mathbf{k}} = \frac{1}{\sqrt{2}}\left[\Delta_{\mathbf{k}}^{\dagger}\left(\frac{3}{2},\frac{3}{2}\middle|\frac{1}{2},\frac{1}{2}\right) - \Delta_{\mathbf{k}}^{\dagger}\left(\frac{3}{2},\frac{-3}{2}\middle|\frac{1}{2},\frac{-1}{2}\right)\right], \tag{80}$$

$$(\tilde{T}_{2(1)}^{\dagger;J=2})_{\mathbf{k}} = \frac{1}{2\sqrt{2}}\left[\Delta_{\mathbf{k}}^{\dagger}\left(\frac{1}{2},\frac{-1}{2}\middle|\frac{3}{2},\frac{3}{2}\right) + \Delta_{\mathbf{k}}^{\dagger}\left(\frac{1}{2},\frac{1}{2}\middle|\frac{3}{2},\frac{-3}{2}\right)\right.$$
$$\left. + \sqrt{3}\left(\Delta_{\mathbf{k}}^{\dagger}\left(\frac{1}{2},\frac{1}{2}\middle|\frac{3}{2},\frac{1}{2}\right) + \Delta_{\mathbf{k}}^{\dagger}\left(\frac{1}{2},\frac{-1}{2}\middle|\frac{3}{2},\frac{-1}{2}\right)\right)\right], \tag{81}$$

$$(\tilde{T}_{2(2)}^{\dagger;J=2})_{\mathbf{k}} = \frac{i}{2\sqrt{2}}\left[\Delta_{\mathbf{k}}^{\dagger}\left(\frac{1}{2},\frac{-1}{2}\middle|\frac{3}{2},\frac{3}{2}\right) - \Delta_{\mathbf{k}}^{\dagger}\left(\frac{1}{2},\frac{1}{2}\middle|\frac{3}{2},\frac{-3}{2}\right)\right.$$
$$\left. + \sqrt{3}\left(\Delta_{\mathbf{k}}^{\dagger}\left(\frac{1}{2},\frac{1}{2}\middle|\frac{3}{2},\frac{1}{2}\right) - \Delta_{\mathbf{k}}^{\dagger}\left(\frac{1}{2},\frac{-1}{2}\middle|\frac{3}{2},\frac{-1}{2}\right)\right)\right], \tag{82}$$

$$(\tilde{T}_{2(3)}^{\dagger;J=2})_{\mathbf{k}} = \frac{1}{\sqrt{2}}\left[\Delta_{\mathbf{k}}^{\dagger}\left(\frac{1}{2},\frac{1}{2}\middle|\frac{3}{2},\frac{3}{2}\right) - \Delta_{\mathbf{k}}^{\dagger}\left(\frac{1}{2},\frac{-1}{2}\middle|\frac{3}{2},\frac{-3}{2}\right)\right]. \tag{83}$$

Odd J: $\frac{3}{2}\otimes\frac{1}{2}$; $[\frac{1}{2}\otimes\frac{3}{2}]$

$$(T_{1(1)}^{\dagger;J=1})_{\mathbf{k}} = \frac{i}{2\sqrt{2}}\left[-\Delta_{\mathbf{k}}^{\dagger}\left(\frac{3}{2},\frac{1}{2}\middle|\frac{1}{2},\frac{1}{2}\right) - \Delta_{\mathbf{k}}^{\dagger}\left(\frac{3}{2},\frac{-1}{2}\middle|\frac{1}{2},\frac{-1}{2}\right)\right.$$
$$\left. + \sqrt{3}\left(\Delta_{\mathbf{k}}^{\dagger}\left(\frac{3}{2},\frac{3}{2}\middle|\frac{1}{2},\frac{-1}{2}\right) + \Delta_{\mathbf{k}}^{\dagger}\left(\frac{3}{2},\frac{-3}{2}\middle|\frac{1}{2},\frac{1}{2}\right)\right)\right], \tag{84}$$

$$(T_{1(2)}^{\dagger;J=1})_{\mathbf{k}} = \frac{1}{2\sqrt{2}}\left[-\Delta_{\mathbf{k}}^{\dagger}\left(\frac{3}{2},\frac{1}{2}\middle|\frac{1}{2},\frac{1}{2}\right) + \Delta_{\mathbf{k}}^{\dagger}\left(\frac{3}{2},\frac{-1}{2}\middle|\frac{1}{2},\frac{-1}{2}\right)\right.$$
$$\left. + \sqrt{3}\left(\Delta_{\mathbf{k}}^{\dagger}\left(\frac{3}{2},\frac{3}{2}\middle|\frac{1}{2},\frac{-1}{2}\right) - \Delta_{\mathbf{k}}^{\dagger}\left(\frac{3}{2},\frac{-3}{2}\middle|\frac{1}{2},\frac{1}{2}\right)\right)\right], \tag{85}$$

$$(T_{1(3)}^{\dagger;J=1})_{\mathbf{k}} = \frac{i}{\sqrt{2}}\left[\Delta_{\mathbf{k}}^{\dagger}\left(\frac{3}{2},\frac{1}{2}\middle|\frac{1}{2},\frac{-1}{2}\right) - \Delta_{\mathbf{k}}^{\dagger}\left(\frac{3}{2},\frac{-1}{2}\middle|\frac{1}{2},\frac{1}{2}\right)\right], \tag{86}$$

$$(\tilde{T}_{1(1)}^{\dagger;J=1})_{\mathbf{k}} = \frac{i}{2\sqrt{2}}\left[\Delta_{\mathbf{k}}^{\dagger}\left(\frac{1}{2},\frac{1}{2}\middle|\frac{3}{2},\frac{1}{2}\right) + \Delta_{\mathbf{k}}^{\dagger}\left(\frac{1}{2},\frac{-1}{2}\middle|\frac{3}{2},\frac{-1}{2}\right)\right.$$
$$\left. - \sqrt{3}\left(\Delta_{\mathbf{k}}^{\dagger}\left(\frac{1}{2},\frac{-1}{2}\middle|\frac{3}{2},\frac{3}{2}\right) + \Delta_{\mathbf{k}}^{\dagger}\left(\frac{1}{2},\frac{1}{2}\middle|\frac{3}{2},\frac{-3}{2}\right)\right)\right], \tag{87}$$

$$(\tilde{T}_{1(2)}^{\dagger;J=1})_{\mathbf{k}} = \frac{1}{2\sqrt{2}}\left[\Delta_{\mathbf{k}}^{\dagger}\left(\frac{1}{2},\frac{1}{2}\middle|\frac{3}{2},\frac{1}{2}\right) - \Delta_{\mathbf{k}}^{\dagger}\left(\frac{1}{2},\frac{-1}{2}\middle|\frac{3}{2},\frac{-1}{2}\right)\right.$$

$$\left. - \sqrt{3}\left(\Delta_{\mathbf{k}}^{\dagger}\left(\frac{1}{2},\frac{-1}{2}\middle|\frac{3}{2},\frac{3}{2}\right) - \Delta_{\mathbf{k}}^{\dagger}\left(\frac{1}{2},\frac{1}{2}\middle|\frac{3}{2},\frac{-3}{2}\right)\right)\right], \tag{88}$$

$$(\tilde{T}_{1(3)}^{\dagger;J=1})_{\mathbf{k}} = \frac{i}{\sqrt{2}}\left[-\Delta_{\mathbf{k}}^{\dagger}\left(\frac{1}{2},\frac{-1}{2}\middle|\frac{3}{2},\frac{1}{2}\right) + \Delta_{\mathbf{k}}^{\dagger}\left(\frac{1}{2},\frac{1}{2}\middle|\frac{3}{2},\frac{-1}{2}\right)\right]. \tag{89}$$

## F Two-channel Kondo interaction derived pairing interactions

The pairing interactions generated from the two-channel Kondo interaction can be classified into even and odd spatial parity, which follows from the even and odd $J$ angular momentum, in Eqs. 91 and 92, respectively,

$$H_{\text{eff}}^{\text{2CK}} = \left[H_{\text{even}}^{\text{2CK}} + H_{\text{odd}}^{\text{2CK}}\right]. \tag{90}$$

For brevity, we drop the superscript indicating the total angular momentum; Fig. 1 indicates the total angular momentum of the Cooper pair.

$$H_{\text{even}}^{\text{2CK}} = -\sum_{\mathbf{k},\mathbf{k}'}\left[-f_1 J_3^2 + 2J_1^2 f_0\right]_{\mathbf{k},\mathbf{k}'} A_{1\mathbf{k}}^{\dagger} A_{1\mathbf{k}'}$$

$$- \sum_{\mathbf{k},\mathbf{k}'}\left[f_1 J_3^2 - 2J_1^2 f_{2\nu}\right]_{\mathbf{k},\mathbf{k}'} E_{(1)\mathbf{k}}^{\dagger} E_{(1)\mathbf{k}'} - \sum_{\mathbf{k},\mathbf{k}'}\left[f_1 J_3^2 + 2J_1^2 f_{2\nu}\right]_{\mathbf{k},\mathbf{k}'} E_{(2)\mathbf{k}}^{\dagger} E_{(2)\mathbf{k}'}$$

$$+ \sum_{\mathbf{k},\mathbf{k}'}\left[2J_1^2 f_{2\mu}\right]_{\mathbf{k},\mathbf{k}'}\left(E_{(1)\mathbf{k}}^{\dagger} E_{(2)\mathbf{k}'} + \text{h.c.}\right) + \sum_{\mathbf{k},\mathbf{k}'}\left[f_1 J_3^2 + 2J_1^2 f_0\right]_{\mathbf{k},\mathbf{k}'}\vec{T}_{2\mathbf{k}}^{\dagger}\cdot\vec{T}_{2\mathbf{k}'}, \tag{91}$$

$$H_{\text{odd}}^{\text{2CK}} = -\sum_{\mathbf{k},\mathbf{k}'}\left[-f_1 J_3^2 - 2J_1^2 f_0\right]_{\mathbf{k},\mathbf{k}'} A_{2\mathbf{k}}^{\dagger} A_{2\mathbf{k}'} - \sum_{\mathbf{k},\mathbf{k}'}\left[-f_1 J_3^2 + 2f_0 J_1^2\right]_{\mathbf{k},\mathbf{k}'}\vec{T}_{-\mathbf{k}}^{\dagger}\cdot\vec{T}_{-\mathbf{k}'} \tag{92}$$

$$- \sum_{\mathbf{k},\mathbf{k}'}\left[f_1 J_3^2 + J_1^2 f_{2\nu} - \sqrt{3}J_1^2 f_{2\mu}\right]_{\mathbf{k},\mathbf{k}'} T_{2(1)\mathbf{k}}^{\dagger} T_{2(1)\mathbf{k}'}$$

$$- \sum_{\mathbf{k},\mathbf{k}'}\left[f_1 J_3^2 - J_1^2 f_{2\nu} + \sqrt{3}J_1^2 f_{2\mu}\right]_{\mathbf{k},\mathbf{k}'} T_{+(1)\mathbf{k}}^{\dagger} T_{+(1)\mathbf{k}'}$$

$$+ \sum_{\mathbf{k},\mathbf{k}'}\left[-\sqrt{3}J_1^2 f_{2\nu} - J_1^2 f_{2\mu}\right]_{\mathbf{k},\mathbf{k}'}\left(T_{2(1)\mathbf{k}}^{\dagger} T_{+(1)\mathbf{k}'} + \text{h.c.}\right)$$

$$- \sum_{\mathbf{k},\mathbf{k}'}\left[f_1 J_3^2 + J_1^2 f_{2\nu} + \sqrt{3}J_1^2 f_{2\mu}\right]_{\mathbf{k},\mathbf{k}'} T_{2(2)\mathbf{k}}^{\dagger} T_{2(2)\mathbf{k}'}$$

$$- \sum_{\mathbf{k},\mathbf{k}'}\left[f_1 J_3^2 - J_1^2 f_{2\nu} - \sqrt{3}J_1^2 f_{2\mu}\right]_{\mathbf{k},\mathbf{k}'} T_{+(2)\mathbf{k}}^{\dagger} T_{+(2)\mathbf{k}'}$$

$$+ \sum_{\mathbf{k},\mathbf{k}'}\left[\sqrt{3}J_1^2 f_{2\nu} - J_1^2 f_{2\mu}\right]_{\mathbf{k},\mathbf{k}'}\left(T_{2(2)\mathbf{k}}^{\dagger} T_{+(2)\mathbf{k}'} + \text{h.c.}\right)$$

$$- \sum_{\mathbf{k},\mathbf{k}'}\left[f_1 J_3^2 - 2J_1^2 f_{2\nu}\right]_{\mathbf{k},\mathbf{k}'} T_{2(3)\mathbf{k}}^{\dagger} T_{2(3)\mathbf{k}'} - \sum_{\mathbf{k},\mathbf{k}'}\left[f_1 J_3^2 + 2J_1^2 f_{2\nu}\right]_{\mathbf{k},\mathbf{k}'} T_{+(3)\mathbf{k}}^{\dagger} T_{+(3)\mathbf{k}'}$$

$$+ \sum_{\mathbf{k},\mathbf{k}'}\left[-2J_1^2 f_{2\mu}\right]_{\mathbf{k},\mathbf{k}'}\left(T_{2(3)\mathbf{k}}^{\dagger} T_{+(3)\mathbf{k}'} + T_{+(3)\mathbf{k}}^{\dagger} T_{2(3)\mathbf{k}'}\right), \tag{93}$$

where the linear combination of the $T_1$ irreps are defined by

$$T_{+(1,3)}^{\dagger} = \frac{1}{\sqrt{5}}\left(-2T_{1(1,3)}^{\dagger;J=1} + T_{1(1,3)}^{\dagger;J=3}\right), \tag{94}$$

$$T^\dagger_{-(1,3)} = \frac{1}{\sqrt{5}}\left(-T^{\dagger;J=1}_{1(1,3)} - 2T^{\dagger;J=3}_{1(1,3)}\right), \tag{95}$$

$$T^\dagger_{+(2)} = \frac{1}{\sqrt{5}}\left(-2T^{\dagger;J=1}_{1(2)} - T^{\dagger;J=3}_{1(2)}\right), \tag{96}$$

$$T^\dagger_{-(2)} = \frac{1}{\sqrt{5}}\left(T^{\dagger;J=1}_{1(2)} - 2T^{\dagger;J=3}_{1(2)}\right). \tag{97}$$

# G  Novel Kondo interaction derived pairing interactions

The pairing interactions arising from the novel Kondo interaction can be classified into two types of models: (i) Cooper pairs formed separately within $j = 1/2$ and $j = 3/2$ scatter off each other, and (ii) Cooper pairs formed with one fermion from $j = 1/2$ and the other from $j = 3/$ sector. We present the Hamiltonians for each of the families of models below.

## G.1  Cooper pairs formed separately within $j = 1/2$ and $j = 3/2$ sectors

The pairing interaction can be decomposed into even and odd parity sectors (which follows from the even and odd $J$ total angular momentum),

$$H^{\text{novel (i)}}_{\text{eff}} = \left[H^{\text{novel (i)}}_{\text{even}} + H^{\text{novel (i)}}_{\text{odd}}\right], \tag{98}$$

where

$$
\begin{aligned}
H^{\text{novel (i)}}_{\text{even}} = &-\sum_{\mathbf{k},\mathbf{k}'}\left[\sqrt{2}f_0 J_2^2\right]_{\mathbf{k},\mathbf{k}'}\left(A^\dagger_{1;j=\frac{3}{2};\mathbf{k}}A_{1;j=\frac{1}{2};\mathbf{k}'} + \text{h.c.}\right) \\
&+\sum_{\mathbf{k},\mathbf{k}'}\left[\sqrt{2}f_{2\nu} J_2^2\right]_{\mathbf{k},\mathbf{k}'}\left(A^\dagger_{1;j=\frac{1}{2};\mathbf{k}}E^{J=2}_{2;\mathbf{k}'} + \text{h.c.}\right) \\
&+\sum_{\mathbf{k},\mathbf{k}'}\left[\sqrt{2}f_{2\mu} J_2^2\right]_{\mathbf{k},\mathbf{k}'}\left(A^\dagger_{1;j=\frac{1}{2};\mathbf{k}}E^{J=2}_{1;\mathbf{k}'} + \text{h.c.}\right) \\
&-\sum_{\mathbf{k},\mathbf{k}'}\left[f_1 J_3^2\right]_{\mathbf{k},\mathbf{k}'}\left(-A^\dagger_{1;j=\frac{3}{2};\mathbf{k}}A_{1;j=\frac{3}{2};\mathbf{k}'} + \vec{E}^{\dagger;J=2}_{\mathbf{k}}\cdot\vec{E}^{J=2}_{\mathbf{k}'}\right). \tag{99}
\end{aligned}
$$

As seen in the above interaction, the $A_1$ irrep arising from the pair formed by $j = 1/2$ sector of conduction electrons has a non-vanishing matrix element with the Cooper pairs formed from the $j = 3/2$ sector; we give the operators the corresponding $j$ sector subscript label. We once again drop the total angular momentum superscript for brevity.

The odd parity pairing interactions are,

$$
\begin{aligned}
H^{\text{novel (i)}}_{\text{odd}} = &-\sum_{\mathbf{k},\mathbf{k}'}\left[\sqrt{2}f_0 J_2^2\right]_{\mathbf{k},\mathbf{k}'}\left(T^\dagger_{1(1);j=1/2;\mathbf{k}}T_{-(1);j=3/2;\mathbf{k}'} + \text{h.c.}\right) \tag{100} \\
&-\sum_{\mathbf{k},\mathbf{k}'}\left[\frac{1}{\sqrt{2}}f_{2\nu}J_2^2 - \sqrt{\frac{3}{2}}f_{2\mu}J_2^2\right]_{\mathbf{k},\mathbf{k}'}\left(T^\dagger_{1(1);j=1/2;\mathbf{k}}T_{+(1);j=3/2;\mathbf{k}'} + \text{h.c.}\right) \\
&-\sum_{\mathbf{k},\mathbf{k}'}\left[\sqrt{\frac{3}{2}}f_{2\nu}J_2^2 + \frac{1}{\sqrt{2}}f_{2\mu}J_2^2\right]_{\mathbf{k},\mathbf{k}'}\left(T^\dagger_{1(1);j=1/2;\mathbf{k}}T_{2(1);j=3/2;\mathbf{k}'} + \text{h.c.}\right) \\
&+\sum_{\mathbf{k},\mathbf{k}'}\left[f_1 J_3^2\right]_{\mathbf{k},\mathbf{k}'}\left(T^\dagger_{-(1);j=3/2;\mathbf{k}}T^\dagger_{-(1);j=3/2;\mathbf{k}'} - T^\dagger_{+(1);j=3/2;\mathbf{k}}T^\dagger_{+(1);j=3/2;\mathbf{k}'}\right.
\end{aligned}
$$

$$- T^\dagger_{2^{(1)};j=3/2;\mathbf{k}} T^\dagger_{2^{(1)};j=3/2;\mathbf{k'}} \Big)$$

$$-\sum_{\mathbf{k},\mathbf{k'}} \Big[ -\sqrt{2} f_0 J_2^2 \Big]_{\mathbf{k},\mathbf{k'}} \Big( T^\dagger_{1^{(2)};j=1/2;\mathbf{k}} T_{-^{(2)};j=3/2;\mathbf{k'}} + \text{h.c.} \Big)$$

$$-\sum_{\mathbf{k},\mathbf{k'}} \Big[ \frac{1}{\sqrt{2}} f_{2\nu} J_2^2 + \sqrt{\frac{3}{2}} f_{2\mu} J_2^2 \Big]_{\mathbf{k},\mathbf{k'}} \Big( T^\dagger_{1^{(2)};j=1/2;\mathbf{k}} T_{+^{(2)};j=3/2;\mathbf{k'}} + \text{h.c.} \Big)$$

$$-\sum_{\mathbf{k},\mathbf{k'}} \Big[ -\sqrt{\frac{3}{2}} f_{2\nu} J_2^2 + \frac{1}{\sqrt{2}} f_{2\mu} J_2^2 \Big]_{\mathbf{k},\mathbf{k'}} \Big( T^\dagger_{1^{(2)};j=1/2;\mathbf{k}} T_{2^{(2)};j=3/2;\mathbf{k'}} + \text{h.c.} \Big)$$

$$+\sum_{\mathbf{k},\mathbf{k'}} \Big[ f_1 J_3^2 \Big]_{\mathbf{k},\mathbf{k'}} \Big( T^\dagger_{-^{(2)};j=3/2;\mathbf{k}} T^\dagger_{-^{(2)};j=3/2;\mathbf{k'}} - T^\dagger_{+^{(2)};j=3/2;\mathbf{k}} T^\dagger_{+^{(2)};j=3/2;\mathbf{k'}}$$

$$- T^\dagger_{2^{(2)};j=3/2;\mathbf{k}} T^\dagger_{2^{(2)};j=3/2;\mathbf{k'}} \Big)$$

$$-\sum_{\mathbf{k},\mathbf{k'}} \Big[ \sqrt{2} f_0 J_2^2 \Big]_{\mathbf{k},\mathbf{k'}} \Big( T^\dagger_{1^{(3)};j=1/2;\mathbf{k}} T_{-^{(3)};j=3/2;\mathbf{k'}} + \text{h.c.} \Big)$$

$$-\sum_{\mathbf{k},\mathbf{k'}} \Big[ -\sqrt{2} f_{2\nu} J_2^2 \Big]_{\mathbf{k},\mathbf{k'}} \Big( T^\dagger_{1^{(3)};j=1/2;\mathbf{k}} T_{+^{(3)};j=3/2;\mathbf{k'}} + \text{h.c.} \Big)$$

$$-\sum_{\mathbf{k},\mathbf{k'}} \Big[ \sqrt{2} f_{2\mu} J_2^2 \Big]_{\mathbf{k},\mathbf{k'}} \Big( T^\dagger_{1^{(3)};j=1/2;\mathbf{k}} T_{2^{(3)};j=3/2;\mathbf{k'}} + \text{h.c.} \Big)$$

$$+\sum_{\mathbf{k},\mathbf{k'}} \Big[ f_1 J_3^2 \Big]_{\mathbf{k},\mathbf{k'}} \Big( T^\dagger_{-^{(3)};j=3/2;\mathbf{k}} T^\dagger_{-^{(3)};j=3/2;\mathbf{k'}} - T^\dagger_{+^{(3)};j=3/2;\mathbf{k}} T^\dagger_{+^{(3)};j=3/2;\mathbf{k'}}$$

$$- T^\dagger_{2^{(3)};j=3/2;\mathbf{k}} T^\dagger_{2^{(3)};j=3/2;\mathbf{k'}} \Big) .$$

The $T_1$ irrep arising from the $j = 1/2$ sector has a non-vanishing matrix element with the Cooper pairs formed from the $j = 3/2$ sector; we once again present the operators with the corresponding $j$ sector subscript label.

## G.2 Cooper pairs formed from $j = 1/2$ and $j = 3/2$ sectors

The interaction arising from this case needs to be treated more carefully as the order in which the fermionic creation operators are written to form a given Cooper pair provides an additional complexity: for example, for the $J = 2$ pair, we can have $E^\dagger_{1\mathbf{k}}$ and $\tilde{E}^\dagger_{1\mathbf{k}}$ presented in Appendix E. To ensure the pairing function is odd under fermion-exchange *and* spatial inversion/parity, we need to consider symmetric and antisymmetric combinations of these pairing operators: one will yield an even under parity combination, while the other will yield and odd under parity combination:

$$\vec{E}^\dagger_{\pm\mathbf{k}} = \frac{1}{\sqrt{2}} \Big( \vec{E}^\dagger_{\mathbf{k}} \pm \vec{\tilde{E}}^\dagger_{\mathbf{k}} \Big), \tag{101}$$

$$\vec{T}^\dagger_{1\pm\mathbf{k}} = \frac{1}{\sqrt{2}} \Big( \vec{\tilde{T}}^\dagger_{1\mathbf{k}} \pm \vec{T}^\dagger_{1\mathbf{k}} \Big), \tag{102}$$

$$\vec{T}^\dagger_{2\pm\mathbf{k}} = \frac{1}{\sqrt{2}} \Big( \vec{\tilde{T}}^\dagger_{2\mathbf{k}} \pm \vec{T}^\dagger_{2\mathbf{k}} \Big). \tag{103}$$

We thus organize the interaction Hamiltonians into even and odd under spatial parity Cooper pairs.

$$H^{\text{novel (ii)}}_{\text{eff}} = H^{\text{novel (ii)}}_{\text{even}} + H^{\text{novel (ii)}}_{\text{odd}}, \tag{104}$$

where

$$H_{\text{even}}^{\text{novel (ii)}} = -\sum_{\mathbf{k},\mathbf{k}'}\Big[f_{2\nu}\beta^2 - f_0\beta^2\Big]_{\mathbf{k},\mathbf{k}'} E_{+(1)\mathbf{k}}^{\dagger} E_{+(1)\mathbf{k}'} - \sum_{\mathbf{k},\mathbf{k}'}\Big[-f_{2\nu}\beta^2 - f_0\beta^2\Big]_{\mathbf{k},\mathbf{k}'} E_{+(2)\mathbf{k}}^{\dagger} E_{+(2)\mathbf{k}'}$$
$$+ \sum_{\mathbf{k},\mathbf{k}'}\Big[-f_{2\mu}\beta^2\Big]_{\mathbf{k},\mathbf{k}'}\Big(E_{+(1)\mathbf{k}}^{\dagger} E_{+(2)\mathbf{k}'} + \text{h.c.}\Big) + \sum_{\mathbf{k},\mathbf{k}'}\Big[f_{2\mu}\beta^2\Big]_{\mathbf{k},\mathbf{k}'}\Big(E_{-(1)\mathbf{k}}^{\dagger} E_{-(2)\mathbf{k}'} + \text{h.c.}\Big)$$
$$- \sum_{\mathbf{k},\mathbf{k}'}\Big[-f_{2\nu}\beta^2 + f_0\beta^2\Big]_{\mathbf{k},\mathbf{k}'} E_{-(1)\mathbf{k}}^{\dagger} E_{-(1)\mathbf{k}'} - \sum_{\mathbf{k},\mathbf{k}'}\Big[f_{2\nu}\beta^2 + f_0\beta^2\Big]_{\mathbf{k},\mathbf{k}'} E_{-(2)\mathbf{k}}^{\dagger} E_{-(2)\mathbf{k}'},$$

(105)

$$H_{\text{odd}}^{\text{novel (ii)}} = -\sum_{\mathbf{k},\mathbf{k}'}\Big[f_0\beta^2 + \tfrac{1}{2}f_{2\nu}\beta^2 - \tfrac{\sqrt{3}}{2}f_{2\mu}\beta^2\Big]_{\mathbf{k},\mathbf{k}'} T_{2+(1)\mathbf{k}}^{\dagger} T_{2+(1)\mathbf{k}'}$$
$$- \sum_{\mathbf{k},\mathbf{k}'}\Big[f_0\beta^2 - \tfrac{1}{2}f_{2\nu}\beta^2 + \tfrac{\sqrt{3}}{2}f_{2\mu}\beta^2\Big]_{\mathbf{k},\mathbf{k}'} T_{1-(1)\mathbf{k}}^{\dagger} T_{1-(1)\mathbf{k}'}$$
$$+ \sum_{\mathbf{k},\mathbf{k}'}\Big[-\tfrac{\sqrt{3}}{2}f_{2\nu}\beta^2 - \tfrac{1}{2}f_{2\mu}\beta^2\Big]_{\mathbf{k},\mathbf{k}'}\Big(-i T_{2+(1)\mathbf{k}}^{\dagger} T_{1-(1)\mathbf{k}'} + \text{h.c.}\Big)$$
$$- \sum_{\mathbf{k},\mathbf{k}'}\Big[f_0\beta^2 + \tfrac{1}{2}f_{2\nu}\beta^2 + \tfrac{\sqrt{3}}{2}f_{2\mu}\beta^2\Big]_{\mathbf{k},\mathbf{k}'} T_{2+(2)\mathbf{k}}^{\dagger} T_{2+(2)\mathbf{k}'}$$
$$- \sum_{\mathbf{k},\mathbf{k}'}\Big[f_0\beta^2 - \tfrac{1}{2}f_{2\nu}\beta^2 - \tfrac{\sqrt{3}}{2}f_{2\mu}\beta^2\Big]_{\mathbf{k},\mathbf{k}'} T_{1-(2)\mathbf{k}}^{\dagger} T_{1-(2)\mathbf{k}'}$$
$$+ \sum_{\mathbf{k},\mathbf{k}'}\Big[-\tfrac{\sqrt{3}}{2}f_{2\nu}\beta^2 + \tfrac{1}{2}f_{2\mu}\beta^2\Big]_{\mathbf{k},\mathbf{k}'}\Big(i T_{2+(2)\mathbf{k}}^{\dagger} T_{1-(2)\mathbf{k}'} + \text{h.c.}\Big)$$
$$- \sum_{\mathbf{k},\mathbf{k}'}\Big[f_0\beta^2 - f_{2\nu}\beta^2\Big]_{\mathbf{k},\mathbf{k}'} T_{2+(3)\mathbf{k}}^{\dagger} T_{2+(3)\mathbf{k}'} - \sum_{\mathbf{k},\mathbf{k}'}\Big[f_0\beta^2 + f_{2\nu}\beta^2\Big]_{\mathbf{k},\mathbf{k}'} T_{1-(3)\mathbf{k}}^{\dagger} T_{1-(3)\mathbf{k}'}$$
$$+ \sum_{\mathbf{k},\mathbf{k}'}\Big[f_{2\mu}\beta^2\Big]_{\mathbf{k},\mathbf{k}'}\Big(-i T_{2+(3)\mathbf{k}}^{\dagger} T_{1-(3)\mathbf{k}'} + \text{h.c.}\Big)$$ (106)
$$- \Big\{(T_{2+}, T_{1-}) \leftrightarrow (T_{2-}, T_{1+})\Big\},$$

where once can notice that the $E_+$ decouples from the $E_-$, and the $T_{2\pm}$ couples to the same component of $T_{1\mp}$.

## H  BCS Mean field Theory Gap equation

The BCS gap equation can be elegantly derived by the Hubbard–Stratonovich (HS) transformation. We present a sketch of the derivation below, focussing on the aspects that require special attention.

We consider the form of a typical collection of interaction terms,

$$S_{\text{int}} = -\int_0^\beta d\tau \sum_{\mathbf{k},\mathbf{k}'}\sum_{\alpha\beta} \overline{X}_{\mathbf{k}\alpha}(\Omega_{\mathbf{k}\mathbf{k}'})_{\alpha\beta} X_{\mathbf{k}'\alpha},$$ (107)

where $A^\dagger, B^\dagger$ are generic Cooper pair operators made up of conduction creation operator bilinears, $\alpha, \beta$ runs over the internal structure of the interaction potential $(\Omega_{\mathbf{k}\mathbf{k}'})_{\alpha\beta}$; as a simple

example, for the pairing instability arising in the $\frac{3}{2} \otimes \frac{3}{2}$ sector's $\vec{E}^{J=2}$ Cooper pairs, $(\Omega_{\mathbf{kk}'})_{\alpha\beta}$ is a $2 \times 2$ matrix in the internal structure, where each entry is a matrix in momentum space.

We now introduce the auxillary field $\vec{z}_{\mathbf{k}}$ into the partition function with its corresponding free action,

$$\int_0^\beta d\tau \sum_{\mathbf{k},\mathbf{k}'} \sum_{\alpha\beta} \overline{z}_{\mathbf{k}\alpha} (V_{\mathbf{kk}'}^g)_{\alpha\beta} z_{\mathbf{k}'\alpha}, \tag{108}$$

where we use the generalized (Moore-Penrose) pseudoinverse. The pseudoinverse is typically introduced when solving a system of linear equations

$$A\vec{x} = \vec{b}. \tag{109}$$

When $A$ is singular matrix, the unique inverse cannot be employed, thus necessitating the introduction of the pseudoinverse which provides the solution to Eq. 109,

$$\vec{x}_* = A^g\vec{b} + (\mathbb{I} - A^g A)\vec{\omega}, \tag{110}$$

where $A^g$ is the pseudoinverse matrix, and $\vec{\omega}$ is an arbitrary vector. Importantly, the pseudoinverse matrix satisfies the identity $AA^gA = A$, and provided that a solution exists for Eq. 109, then

$$AA^g\vec{b} = \vec{b}, \tag{111}$$

where we emphasize for the pseudoinverse matrix, $A^gA \neq \mathbb{I}$ in general.

Returning back to the path integral, we perform the shift of the auxillary field,

$$z_{\mathbf{k}\alpha} = \Delta_{\mathbf{k}\alpha} + \sum_{\mathbf{p};\gamma} (V_{\mathbf{kp}})_{\alpha\gamma} X_{\mathbf{p}\gamma}, \tag{112}$$

$$\overline{z}_{\mathbf{k}\alpha} = \overline{\Delta}_{\mathbf{k};\alpha} + \sum_{\mathbf{p},\gamma} \overline{X}_{\mathbf{p}\gamma} (V_{\mathbf{pk}})_{\gamma\alpha}, \tag{113}$$

where we now introduce the superconducting HS field $\Delta_{\mathbf{k}\alpha}$. To make progress, the HS field is taken to satisfy the criterion,

$$\sum_{\mathbf{kk}'} \sum_{\alpha\beta} (V_{\mathbf{pk}})_{\gamma\alpha} (V_{\mathbf{kk}'}^g)_{\alpha\beta} \Delta_{\mathbf{k}'\beta} = \Delta_{\mathbf{p}\gamma}. \tag{114}$$

For an invertible interaction, Eq. 114 reduces to a trivially true statement, but for a non-invertible matrix this equality is rationalized as the application of Eq. 111 with the identification of $\vec{b}$ as the static HS field satisfying the linear equation $\sum_{\mathbf{k}'} \sum_\beta (V_{\mathbf{kk}'})_{\alpha\beta} \langle X_{\mathbf{k}'\beta} \rangle = \Delta_{\mathbf{k}\alpha}$. We thus we arrive at the effective interaction,

$$H_{\mathrm{eff}} = \sum_{\mathbf{k},\mathbf{k}'} \sum_{\alpha,\beta} \overline{\Delta}_{\mathbf{k}\alpha} (V_{\mathbf{kk}'}^g)_{\alpha\beta} \Delta_{\mathbf{k}'\beta} + \sum_{\mathbf{k};\alpha} (\overline{\Delta}_{\mathbf{k}\alpha} X_{\mathbf{k}\alpha} + \overline{X}_{\mathbf{k}\alpha} \Delta_{\mathbf{k}\alpha}), \tag{115}$$

with the accompanying path integral,

$$Z = \int \mathcal{D}[\overline{\Delta}, \Delta] \mathcal{D}[\overline{c}, c] e^{-\int_0^\beta d\tau (\sum_{\mathbf{k}} \vec{\overline{c}}_{\mathbf{k}} (\partial_\tau + \epsilon_{\mathbf{k}}) \vec{c}_{\mathbf{k}} + H_{\mathrm{eff}})}, \tag{116}$$

where we reinserted the free fermion action, where $\epsilon_{\mathbf{k}} = \frac{\mathbf{k}^2}{2m} - \mu_F$.

To make progress, the fermions are re-written in a Nambu basis depending on the nature of the interactions in Eq. 115. For the Cooper pairs arising in the $\frac{3}{2} \otimes \frac{3}{2}$ sector, the Nambu basis is,

$$
\vec{\psi}^{\dagger}_{\mathbf{k};\frac{3}{2}\otimes\frac{3}{2}} =
\begin{bmatrix}
c^{\dagger}_{\frac{3}{2},\frac{2}{2};\mathbf{k}} \\
c^{\dagger}_{\frac{3}{2},\frac{1}{2};\mathbf{k}} \\
c^{\dagger}_{\frac{3}{2},\frac{-1}{2};\mathbf{k}} \\
c^{\dagger}_{\frac{3}{2},\frac{-3}{2};\mathbf{k}} \\
c_{\frac{3}{2},\frac{2}{2};-\mathbf{k}} \\
c_{\frac{3}{2},\frac{1}{2};-\mathbf{k}} \\
c_{\frac{3}{2},\frac{-1}{2};-\mathbf{k}} \\
c_{\frac{3}{2},\frac{-3}{2};-\mathbf{k}}
\end{bmatrix} ,
\tag{117}
$$

and for models that involve both $j = \frac{3}{2}$ and $j = \frac{1}{2}$ fermions the basis is expanded to be

$$
\vec{\psi}^{\dagger}_{\mathbf{k};\frac{1}{2}\otimes\frac{3}{2}} =
\begin{bmatrix}
c^{\dagger}_{\frac{3}{2},\frac{2}{2};\mathbf{k}} \\
c^{\dagger}_{\frac{3}{2},\frac{1}{2};\mathbf{k}} \\
c^{\dagger}_{\frac{3}{2},\frac{-1}{2};\mathbf{k}} \\
c^{\dagger}_{\frac{3}{2},\frac{-3}{2};\mathbf{k}} \\
c^{\dagger}_{\frac{1}{2},\frac{1}{2};\mathbf{k}} \\
c^{\dagger}_{\frac{1}{2},\frac{-1}{2};\mathbf{k}} \\
c_{\frac{3}{2},\frac{2}{2};-\mathbf{k}} \\
c_{\frac{3}{2},\frac{1}{2};-\mathbf{k}} \\
c_{\frac{3}{2},\frac{-1}{2};-\mathbf{k}} \\
c_{\frac{3}{2},\frac{-3}{2};-\mathbf{k}} \\
c_{\frac{1}{2},\frac{1}{2};-\mathbf{k}} \\
c_{\frac{1}{2},\frac{-1}{2};-\mathbf{k}}
\end{bmatrix} .
\tag{118}
$$

In terms of the Nambu basis, the HS transformed path integral is bilinear and as such we can simply integrate out the fermionic fields. The subsequent fermionic determinant is of the form $\Pi_{i=1,\ldots,m}(\omega_n^2 + E_{i\mathbf{k}}^2)^2$, where $\omega_n$ is the fermionic Matsubara frequency, $m$ runs over the possible quasiparticle energies. In the cases we consider in this work, $m = 1, 2, 3$ are the only possibilities; for $m = 1$, the determinant is $(\omega_n^2 + E_{1\mathbf{k}}^2)^4$. Finally, approximating the path integral via the standard saddle-point approximation, leads to the mean-field free energy,

$$
\begin{aligned}
F_{\text{MFT}} &= -T \ln Z_{BCS} \\
&= -2T \sum_{\mathbf{k},\omega_n} \sum_{i=1\ldots m} \ln(\omega_n^2 + E_{i\mathbf{k}}^2) + \beta T \sum_{\mathbf{k},\mathbf{k}'} \sum_{\alpha,\beta} \overline{\Delta}_{\mathbf{k}\alpha} (\Omega^g_{\mathbf{k}\mathbf{k}'})_{\alpha\beta} \Delta_{\mathbf{k}'\beta} ,
\end{aligned}
\tag{119}
$$

where $\beta = 1/T$ in the second term comes from integrating a static term over $\tau \in [0, \beta]$. Extremizing the mean-field free energy with respect to $\overline{\Delta_{\mathbf{q}\gamma}}$ and multiplying by $\sum_{\mathbf{q}} \sum_{\gamma} (\Omega_{\mathbf{pq}})_{\alpha\gamma}$ leads to Eq. 30 in the main text. As mentioned above, for the case of a single quasiparticle energy ($m = 1$) the right hand side of Eq. 30 has an additional factor of 2.

In solving the mean-field equations, we employ the following choices for the phenomenological constants: $m = 0.2, \mu_F = 1.0, a_0 = 2.5, a_1 = 0.67, a_2 = 1.67, m_{\mathcal{Q}} = 4.0, m_{\mathcal{O}} = 20.8$. The values of phenomenological Landau values were chosen so as to connect with the experimental setting. In particular, the quadrupolar mass term ($m_{\mathcal{Q}} \sim (T - T_{\mathcal{Q}})$) was taken to be

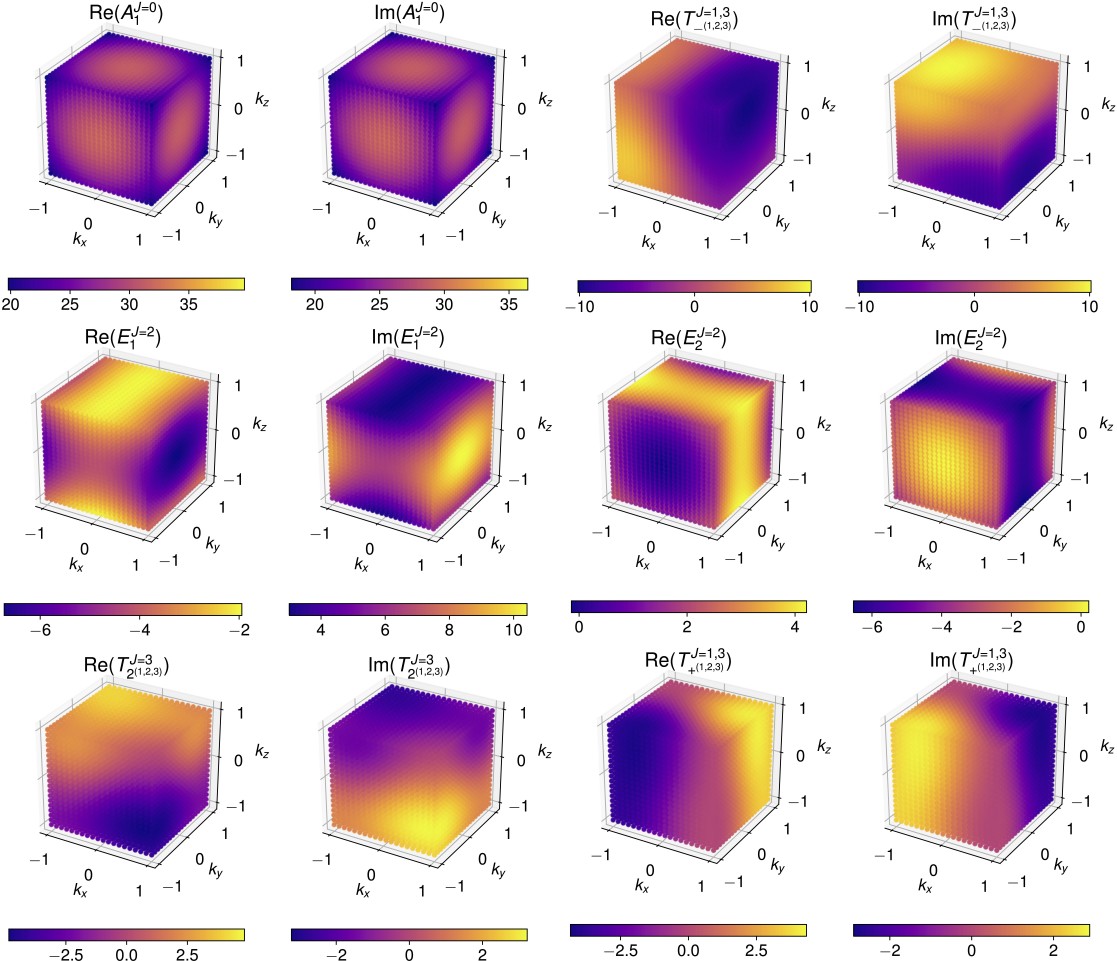

Figure 5: Momentum space distribution of the superconducting states arising from the two-channel Kondo interaction presented in Table 1 and Fig. 1.

smaller than the octupolar mass ($m_{\mathcal{O}} \sim (T - T_{\mathcal{O}})$) to reflect the fact that the proposed octupolar ordering temperature $T_{\mathcal{O}}$ is lower than the quadrupolar ordering temperature $T_{\mathcal{Q}}$ [65]; here $T$ is the temperature, which is taken to be above $T_{\mathcal{Q},\mathcal{O}}$ in the paramagnetic phase. The choices of the other phenomenological values $a_{0,1,2} > 0$ were chosen to ensure that the multipolar action is non-singular, thus permitting the Gaussian fluctuations to be integrated out. For the multipolar Kondo interactions, we choose uniformly $\alpha, \beta, \gamma = 10$, where in the two-channel (novel) interaction $\beta = 0$ ($\alpha = 0$).

# I   Momentum space distribution of superconducting states

We present in Fig. 5, 6, 7, 8, 9 the momentum space distribution of the realized superconducting states induced by the two-channel and novel Kondo interactions. For the cases where there exist multiple degenerate solutions, we present the distribution of one solution for clarity.

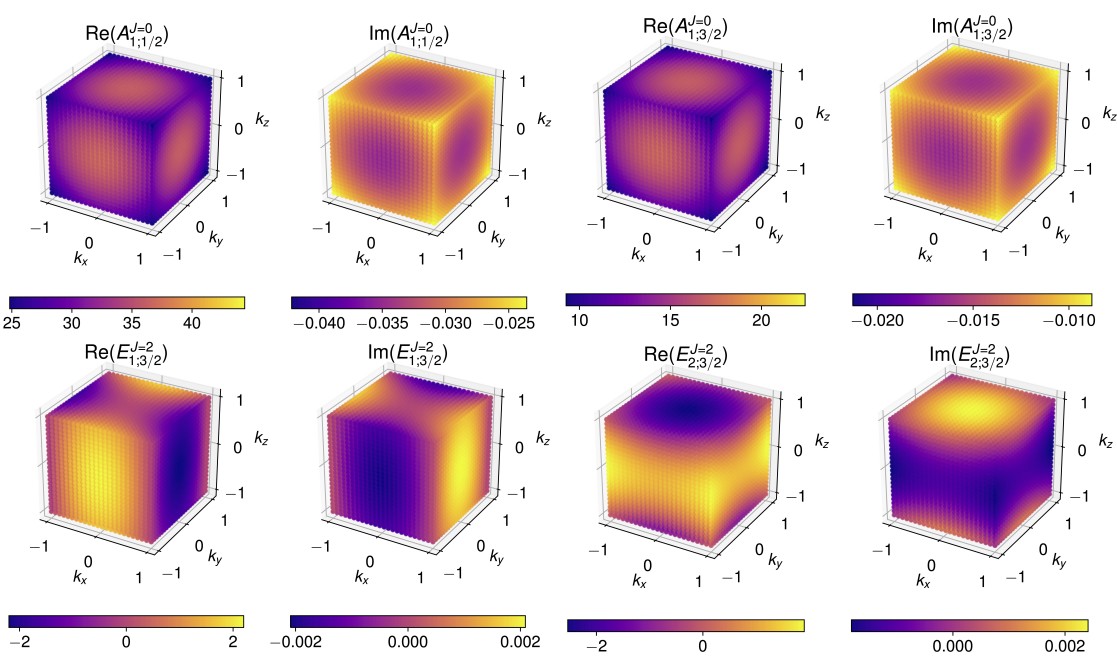

Figure 6: Momentum space distribution of the superconducting states arising from the novel Kondo interaction of even parity presented in Table 2 and Fig. 2.

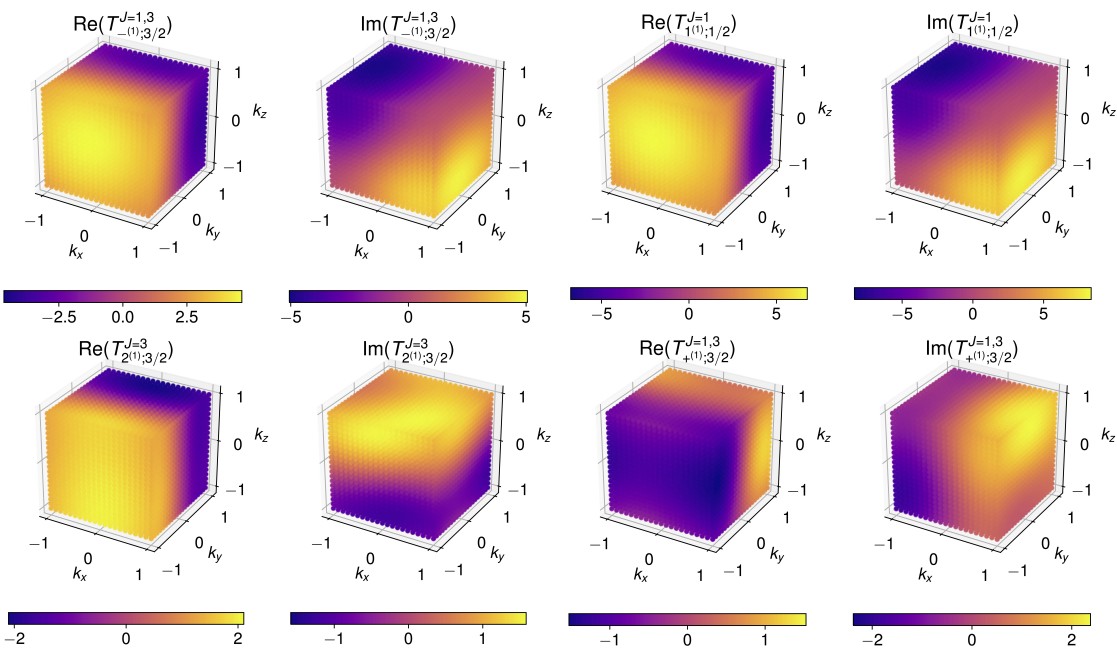

Figure 7: Momentum space distribution of the superconducting states ($j = 1$ component) arising from the novel Kondo interaction presented in Table 2 and Fig. 2.

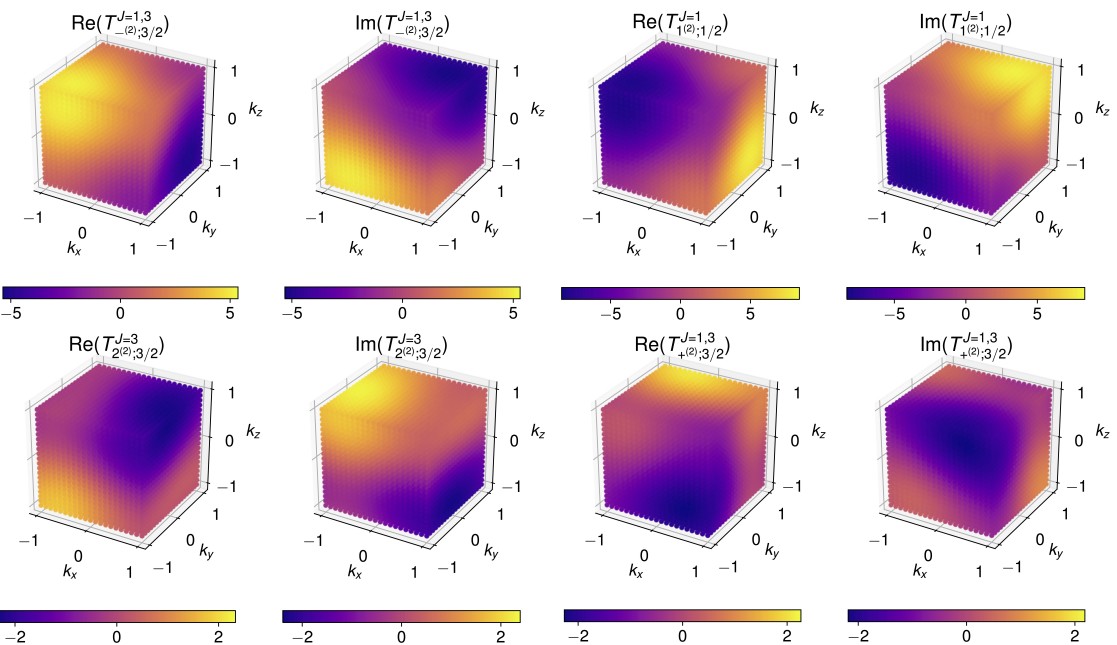

Figure 8: Momentum space distribution of the superconducting states ($j = 2$ component) arising from the novel Kondo interaction presented in Table 2 and Fig. 2.

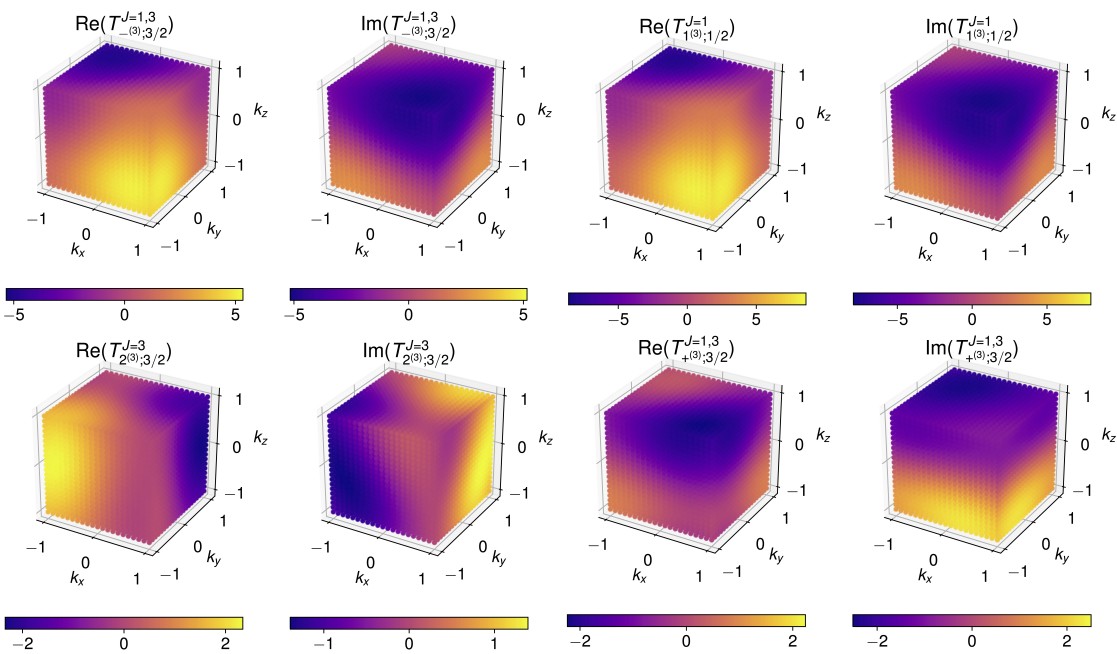

Figure 9: Momentum space distribution of the superconducting states ($j = 3$ component) arising from the novel Kondo interaction presented in Table 2 and Fig. 2.

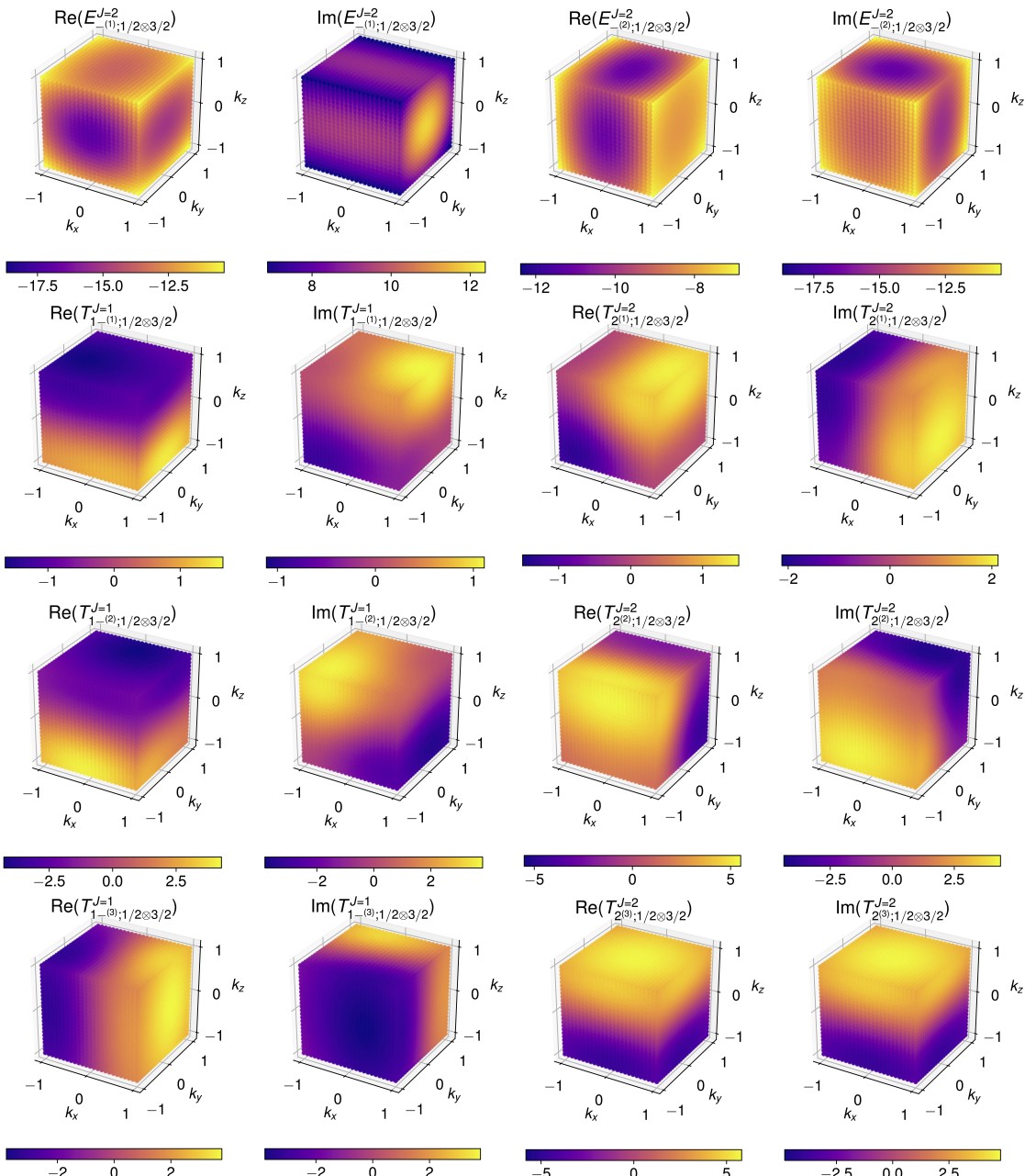

Figure 10: Momentum space distribution of the superconducting states arising from the novel Kondo interaction presented in Table 2 and Fig. 3.

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
