# Peer review of "Unconventional Superconductivity arising from Multipolar Kondo Interactions"

_SciPost Physics, doi:SciPost Phys. 12, 057 (2022)_

## Round 1 · Referee Report · Anonymous · 2021-5-24

Report
In the manuscript titled “Unconventional Superconductivity arising from Multipolar Kondo Interactions”, the authors study higher-angular momentum Cooper pairs generated by two-channel Kondo and novel multipolar Kondo interactions, which are based on their previous studies of the multipolar Kondo effect. This theoretical model applies to the superconductivity found in strongly correlated non-Kramers systems such as Pr$T_2X_{20}$ ($T$ = transition metals, $X$ = Zn, Al). This work is a comprehensive study with new physical insights, and therefore I recommend this manuscript for publication in SciPost Physics after minor revision. Below are my comments on the scientific content and the writing style.
[1] In Sec. 3 and Appendix C, the authors describe the electron-electron interactions from multipolar Kondo effects. However, the assumed condition to obtain the attractive force for the superconductivity is unclear. The authors should clarify such condition.
[2] In Fig.4, the authors show gapless nodes for the odd-parity order parameters. However, it is still difficult for readers to visualize. The authors should revise it, for example, by showing the Bogoliubov Fermi surface. Besides, the labels for the subfigures (a), (b) etc. should be added.
[3] In Sec. 6, the authors describe the properties of the relevant superconducting state found in this research. In my view, it is helpful for readers if you can include some description about the topological feature and whether time reversal symmetry is preserved or not for the relevant superconducting states.
[4] The authors use the point-group irrep to classify the pairing functions. I understand this is theoretically rigorous for the spin-orbit coupled system, but I think such notation is not accessible for experimentalists and non-experts. I would recommend the authors to provide the relation of the point-group irrep to the conventional notation such as $p$-wave or $d$-wave with some figures illustrating the order parameters in $k$-space.
[5] There are some errors in the reference list; Ref. [12] and [13] are duplicated, and there are stylistic errors in the publication titles (for example, URu$_2$Si$_2$, UPt$_3$, UNi$_2$Al$_3$, Knight shift, etc. are not capitalized). Authors should check carefully and revise them.
Author: Adarsh Patri on 2021-07-16 [id 1576]
(in reply to Report 1 on 2021-05-24)
Dear Referee,
We thank you for your insightful questions and comments.
We are attaching a detailed PDF that addresses the questions you raised.
Sincerely,
Adarsh S. Patri and Yong Baek Kim
Author: Adarsh Patri on 2021-07-16 [id 1575]
(in reply to Report 3 on 2021-06-01)Dear Referee,
We thank you for your insightful questions and comments.
We are attaching a detailed PDF that addresses the questions you raised.
Sincerely,
Adarsh S. Patri and Yong-Baek Kim
Attachment:
ref_report_sc_multipolar.pdf

---

## Round 1 · Referee Report · Anonymous · 2021-6-1

Strengths
1. The paper is clearly written for the most part, and addresses the experimentally relevant question of what novel types of pairing may be present due to multipolar fluctuations.
2. The calculation is rigorously done and carefully explained in the appendices.
Report
The paper is well written and addresses the experimentally relevant question of what novel types of pairing might be present due to multipolar fluctuations, and what the gap structure might be. It is potentially relevant to some Pr-based superconductors with multi-polar order. The calculation appears to be rigorously done and details are shown in the appendices.
While I think that the authors calculations are sound, the interpretation seems incorrect, or at least poorly explained. The symmetry aspects here are not clear. The authors claim that superconducting order parameters (OPs) in different irreducible representations coexist, but it is really not clear what they mean. Do they mean something like s+id pairing, where the OPs couple quadratically? They discuss the Cooper pairs scattering off of one another, which suggests a quadratic coupling. However, the language in the main paper, and the appendices (particularly F and G) suggest a linear coupling. If that is the case, these are not different irreps at all, and the effect of the different symmetrizing/antisymmetrizing, spin-orbit coupling terms has just not been properly accounted for. I would like to see a Ginzburg-Landau theory for the superconducting order parameters, with the final irrep of the order parameter given. Given that the high temperature state is paraquadrupolar, there should be no linear couplings between distinct irreps allowed.
Given the authors are examining these higher order J pairings and their parity, the time-reversal properties should also be discussed.
Fig. 3 is the most confusing, as they claim not only to be mixing irreps, but also to be mixing time-reversal odd and even order parameters (odd and even J). Again, I don't see how these can mix if the high temperature state is not breaking time-reversal, and particularly given that the OPs have a fixed parity. Inversion symmetry is not broken, and all the order parameters appear to be even-frequency. This behavior needs to be explained in detail, ideally with a Ginzburg-Landau theory.
The role of symmetrization or antisymmetrization needs to be discussed carefully. It appears to be playing the role of another degree of freedom that comes into the overall antisymmetric nature of the electron wavefunction.
At the moment, I cannot recommend publication until these serious issues are resolved.

---

## Round 2 · Referee Report · Anonymous (Referee 2) · 2021-8-17

Report
I am also concerned by their derivation of this q-dependent interaction vertex. Typically, if I write down a pairing interaction with f(q = k-k’) = cos qx - cos qy, then I separate it into even/odd parity components with f(k-k’) +/- f(k+k’) and then separate these to write f(k) f(k’), where then the d-wave symmetry of f(q) is actually split between the two superconducting order parameters, as opposed to keeping it with the interaction vertex, as they do here. Now, given the symmetry of the Gamma’s involved, they may be correct to do what they have done, however, it is at all not obvious. Given that what they are doing is so unusual, they must explain why their choice is the correct one, in detail, particularly why the momentum dependence is being kept with the vertex rather than going with the superconducting order parameter.
These are serious concerns, and I cannot support publication until they are addressed.

Author: Adarsh Patri on 2021-10-29 [id 1888]
(in reply to Report 2 on 2021-08-17)Dear Referee,
We thank you for your insightful questions, and we attach a PDF document answering your questions.
We have also incorporated the respective changes in the revised manuscript.
Sincerely,
Adarsh S. Patri, Yong Baek Kim
Attachment:
ref_report_sc_multipolar_2.pdf

---

## Round 2 · Author Response

Thank you for the insightful questions regarding our manuscript.
We have answered all the questions and implemented the changes recommended by referees in our revised manuscript.
We have attached a detailed reply to the report (PDF format) under the author reply to the referee for convenience.
Thank you.

---

## Round 2 · List of Changes

The specific additions to the manuscript are (also detailed in the PDF of the author reply to the referee): - Described the physical reasoning behind the choices of the phenomenological parameters (in Appendix H). - Included a revised caption and labelled subfigures in Fig. 4 for clarity. - Described the time-reversal properties of the realized superconducting states from both two-channel and novel Kondo interactions. - Provided the momentum-space distribution of the various order parameters (to make connections with conventional discussions of superconductivity) in a newly constructed Appendix I. - Corrected stylistic errors in the References. - Clarified and confirmed the anti-symmetrization of the Cooper pair operators.
The additions are indicated in blue font in the revised manuscript for convenience.

---

## Round 3 · Referee Report · Anonymous · 2021-11-30

Report
I am satisfied with the changes in the discussion about the momentum dependence. The discussion about “coupled” order parameters is now just absent from the text, with no explanation of what it means to have coupled order parameters (it sounds like this means they were found together in solving the coupled BCS equations for some sets of parameters). Some brief discussion of this should be added somewhere (perhaps on page 9). After this change, I think the paper is okay to be published.
Requested changes
1. Add some brief discussion on page 9 (or elsewhere) of what it means to have "coupled" order parameters.
Author: Adarsh Patri on 2021-12-01 [id 1998]
(in reply to Report 1 on 2021-11-30)Dear Reviewer,
We thank you for your comment and the suggested revision.
We have revised our manuscript to discuss the existence of the multiple order parameter solutions (found by solving the coupled BCS equations) at the bottom of Page 10.
Sincerely,
Adarsh S. Patri and Yong-Baek Kim

---

## Editorial Decision

published